



# Numerical investigation on spectral geometries and their relation to non-Gaussianity in sea states with occurrence of rogue waves: wind-sea dominated events

Xingjie Jiang[1], Tingting Zhang[1,2], Dalu Gao[1], Daolong Wang[1,2]

[1]First Institute of Oceanography (FIO), Ministry of Natural Resources (MNR), Qingdao, 266061, China
[2]Ocean University of China, Qingdao, 266071, China

*Correspondence to*: Xingjie Jiang (jiangxj@fio.org.cn)

**Abstract.** The occurrence of rogue waves is closely related to the non-Gaussianity of sea states, and the non-Gaussianity is sensitive to the combination of three spectral geometries: wave steepness, bandwidth, and directional spreading. This paper presents a set of non-Gaussianity references that allow quantitative comparison of the non-Gaussianity of sea states with various combinations of the three geometries. In addition, an approach to introduce arbitrary 2D wave spectra into the references is presented, which allows quantitative investigation of the non-Gaussianity and the corresponding geometries in given sea states. Application in relation to certain rogue waves that occurred in wind-sea dominated sea states showed that the non-Gaussianity of skewness presented high values when those events occurred. However, abnormal values of kurtosis could not be found within the same period, indicating that third-order modulational instabilities were inactive in those events. Quantitative analyses based on the newly presented references revealed that the rogue waves that occurred in wind-sea dominated sea states, and presented extreme height and extreme destructive power, could hardly be formed from the modulational instabilities. This was because of not only the broad energy distribution in terms of direction, but also the broad bandwidth attributable to the developed wind-sea state.

## 1 Introduction

Rogue/freak/extreme waves are highly destructive ocean waves that represent serious threat to various marine activities, e.g., sea voyages, ocean fishing, and oil exploitation. Several physical mechanisms have been proposed to explain the formation of such waves (Kharif et al., 2009; Kharif and Pelinovsky, 2003), including both linear and nonlinear theories. In explaining the occurrence of rogue waves in open seas, it has been suggested that the nonlinear mechanisms that relate to the second- and third-order nonlinear wave–wave interactions appear most reasonable (Fedele, 2008; Fedele and Tayfun, 2009; Janssen, 2003).





The nonlinear wave-energy focusing caused by these nonlinearities can lead to the formation of rogue waves, but it causes the statistics of wave surface elevations to deviate from the Gaussian (normal) distribution, resulting in non-Gaussian sea states (Longuet-Higgins, 1963). Commonly used measures of the non-Gaussianity of sea state are skewness $\mu_3$ and (excess) kurtosis $\mu_4$:

$$\mu_3 = \frac{\langle \eta^3 \rangle}{\langle \eta^2 \rangle^{3/2}}, \mu_4 = \frac{\langle \eta^4 \rangle}{\langle \eta^2 \rangle^2} - 3, \tag{1}$$

where $\eta$ denotes the wave surface elevation and the terms in angled brackets denote statistical averages. It is clear that the skewness is contributed entirely by the second-order nonlinear interactions between bound waves (Fedele and Tayfun, 2009; Janssen, 2009; Tayfun, 1980; Tayfun and Fedele, 2007). The kurtosis comprises a dynamic component ($\mu_4^{free}$) due to third-order quasi-resonant interactions (Janssen, 2003; Mori and Janssen, 2006) between free waves and another bound component ($\mu_4^{bound}$) induced by both second- and third-order bound-wave nonlinearities (Fedele, 2008; Fedele and Tayfun, 2009;
Janssen and Bidlot, 2009; Tayfun, 1980; Tayfun and Lo, 1990), which can be written as follows:

$$\mu_3 = \mu_3^{bound}, \mu_4 = \mu_4^{free} + \mu_4^{bound}. \tag{2}$$

The non-Gaussianity of sea state is also sensitive to the geometries of the corresponding wave spectrum, and the relation has been well established using theoretical models, e.g., Janssen (2003,2009) and Fedele and Tayfun (2009), and confirmed by
laboratory/numerical experiments, e.g., Onorato et al. (2009a,b), Toffoli et al. (2009), Waseda et al. (2009), and Fedele (2015). Generally, at least three geometries, i.e., wave steepness (hereafter, SP), bandwidth (BW), and directional spreading (DS) can influence skewness, or kurtosis, or both. For steeper SP and narrower BW and DS, it can be concluded that the deviation from Gaussianity will be greater, resulting in higher probability of the occurrence of a rogue wave in such a sea state.


The trends of change of skewness/kurtosis with SP, BW, and DS have been well studied, and the parameters/indicators of skewness and kurtosis can also be obtained through various expressions containing the three geometries (Barbariol et al., 2015; Fedele, 2016; Fedele et al., 2012; Janssen, 2017; Janssen and Bidlot, 2009). However, it remains difficult to assess directly the severity of the deviation from Gaussianity for a given sea state, or to infer the nature of the dominant
nonlinearities. For example, it is known that kurtosis is closely related to third-order quasi-resonant interactions that might cause modulational instabilities (MI), and that large values of kurtosis can be observed in relation to sea states with extremely narrow DS (long-crested sea states), indicating active MI in such conditions. However, despite existing theories or the skewness/kurtosis parameters obtained via the expressions mentioned above, the magnitude of the kurtosis or the narrowness of the DS required for triggering the MI remains unclear. Moreover, the question of whether there are any other



thresholds for the SP and BW geometries remains to be resolved. This is attributable entirely to the lack of a set of operational references with which to compare the non-Gaussianity parameters/indicators obtained with various combinations of the three geometries.

It should also be noted that the skewness/kurtosis parameters mentioned above are prepared for inclusion in certain
wave/crest height probability distribution functions that are intended to obtain the exceptional maximum wave height in rogue wave forecasting systems, e.g., the ECMWF-IFS WAM (ECMWF, 2016; Janssen, 2017; Janssen and Bidlot, 2009) and the Space–Time Extremes forecasting included in version 5.16 of the WWIII (Barbariol et al., 2015, 2017; The WAVEWATCH III R Development Group, 2016). The non-Gaussianity parameters, especially dynamic kurtosis, were first obtained from theoretical models derived under the narrowband assumption in an environment with unidirectional wave
propagation. Therefore, it is necessary to conduct calibrations of those parameters for spectra obtained in real oceans. Such calibrations, undertaken according to the performance of the final forecasting results, involve additional complicated factors that might influence estimation of the non-Gaussianity via the spectral geometries in given sea states.

In this study, we focused solely on the relation of the spectral geometries of SP, BW, and DS to the skewness and kurtosis in
open sea states, and only second- and third-order nonlinear wave–wave interactions were considered in the relationship. Numerous numerical experiments were performed based on the High-Order Spectral Method (HOSM) (Dommermuth and Yue, 1987; West et al., 1987). In those HOSM simulations, various 2D spectra were adopted as initial conditions, and indicators of skewness and kurtosis could be obtained from the simulated non-Gaussian wave fields. Then, the three geometries of the initial spectra were connected to the derived indicators of skewness and kurtosis. Those connections
constituted quantitative references for determining the non-Gaussianity of the sea states corresponding to different combinations of the three geometries. The newly constituted references could also be applied to arbitrary 2D wave spectra with a newly developed approach. Investigation of certain selected real rogue waves was performed based on the newly proposed references. The remainder of this paper is organized as follows. The establishment of the non-Gaussianity references and the behaviour of the non-Gaussianity indicators corresponding to various combinations of the three
geometries are elucidated in Sect. 2. Application to rogue wave events and related analyses are described in Sect. 3. Finally, the derived conclusions and a discussion are presented in Sect. 4.

## 2   Establishment of the non-Gaussianity references

### 2.1   Experimental environment based on HOSM

Skewness and kurtosis are high-order statistical characteristics of wave surface elevations. The evolution of the surface
elevations can be modelled using numerical integration of the potential Euler equations. Assuming a fluid is inviscid and incompressible and the flow irrotational, the continuity equation reduces to the Laplace equation for the velocity potential $\phi$:





$$\nabla^2 \phi + \frac{\partial^2 \phi}{\partial z^2} = 0,$$

and the surface elevations $\eta$ can be obtained by solving the Laplace equation for $\phi$ on surface level $z = \eta$, with the following:

$$\frac{\partial \eta}{\partial t} = (1 + |\nabla \eta|^2)W - \nabla \tilde{\phi} \cdot \nabla \eta$$


$$\frac{\partial \tilde{\phi}}{\partial t} = -g\eta - \frac{1}{2}|\nabla \tilde{\phi}|^2 + \frac{1}{2}(1 + |\nabla \eta|^2)W^2,$$

where $\tilde{\phi} = \phi(\vec{x}, z = \eta(\vec{x}, t), t)$ and $W(\vec{x}, t) = \frac{\partial \tilde{\phi}}{\partial z}\Big|_{z=\eta(\vec{x},t)}$. Using a numerical algorithm called the High-Order Spectral Method (Dommermuth and Yue, 1987; West et al., 1987), $\phi$ can be expanded to a prescribed order $M$, which transforms the complicated Dirichlet problem for $\phi$ on the level of $z = \eta$ into $M$ simpler Dirichlet problems for $\phi^{(m)}$ on $z = 0$. The HOSM can represent the solving of $\phi$ with high accuracy and acceptable efficiency. A number of previous studies related to open

wave field nonlinearities have been performed with HOSM, e.g., nonlinear energy transfers (Tanaka, 2001), bimodal seas (Onorato et al., 2010), and rogue waves (Bitner-Gregersen et al., 2014; Bitner-Gregersen and Toffoli, 2014; Toffoli et al., 2008a, 2008b, 2009, 2010; Xiao et al., 2013), which are most relevant to this study.

In this study, the HOSM experimental environment was established based on the open-source software package HOS-ocean
ver.1.5 (Ducrozet et al., 2016), which was developed at the Laboratoire de recherche en Hydrodynamique, Énergétique et Environnement Atmosphérique of the École Centrale de Nantes (France). HOS-ocean has been validated extensively in terms of nonlinear regular wave propagation (Bonnefoy et al., 2010), and it has also been adopted widely in previous research on the modelling of rogue waves, e.g., Ducrozet et al. (2007), Ducrozet and Gouin (2007), and Jiang et al. (2019). HOS-Ocean can ensure the stability and convergence of the calculation. For example, all aliasing errors generated in the
nonlinear terms can be removed. The time integration is performed by means of an efficient fourth-order Runge–Kutta Cash–Karp scheme, and the time step can be selected automatically to a desired level of accuracy (or so-called tolerance, whose typical values are in the range $[10^{-5}, 10^{-7}]$; in this study, the accuracy achieved was $10^{-7}$). Moreover, a relaxation period of $10T_p$ (where $T_p$ denotes the peak wave period) together with a relevant parameter ($n = 4$) have been considered to remove numerical instabilities attributable to fully nonlinear computations that start with linear initial conditions
(Dommermuth, 2000). Additional details can be found in Ducrozet et al. (2016).

The HOSM experimental environment considers a physical space of size $L_x = xlen \times \lambda_p$ and $L_y = ylen \times \lambda_p$, where $xlen = ylen = 25.5$, $\lambda_p$ is the wavelength at the peak frequency, and the discretization of the space is set as $N_x \times N_y = 256 \times 256$. As HOSM is a pseudo-spectral method, the spectral resolution of the pseudo-spectra can then be determined as
$\Delta k_{x,y} = 2\pi/L_{x,y}$, and the spectral space extends from zero to $k_{max} = \frac{N_{x,y}-1}{2} \times \Delta k$, where $k_{max} = 5k_p$ is the cut-off frequency. As HOSM cannot deal with wave breaking issues, the simulations adopted a typical cut-off frequency $k_{max} =$


$5k_p$, which allows accurate solution of the most energetic part of the spectrum and restricts the breaking of waves in the wave field to within a very limited level (Ducrozet et al., 2017). For both the physical and the spectral space, the $x$-direction was taken as the principal direction of the pseudo-spectra, the $y$-direction was vertical to the $x$-direction, and for the $z$-

direction in the physical space, infinite water depth was adopted.

As the known influence of spectral geometries on non-Gaussianity can be achieved only through wave–wave nonlinearities, the HOSM simulations considered only the nonlinearities and ignored other factors that might influence the non-Gaussianity of the simulated wave fields, e.g., the wind force and energy dissipation due to breaking. It is known that the non-

Gaussianity of sea state is related only to the second- and third-order nonlinearities; therefore, nonlinearities up to the third order (i.e., M = 3) were considered in the simulations.

## 2.2  Initial conditions

To ensure the initial conditions of the HOSM simulations were close to real sea states, the JONSWAP spectrum and specific directional spreading were introduced. The JONSWAP spectrum (Hasselmann et al., 1973) can be expressed as follows:

$$S(f) = B\left(\frac{H_s}{4}\right)^2 \frac{f_p^4}{f^5}\exp\left[-1.25\left(\frac{f}{f_p}\right)^{-4}\right]\gamma^{\exp\left[-\frac{1}{2\beta^2}\left(\frac{f}{f_p}-1\right)^2\right]}, \tag{3}$$

where $H_s$ is the significant wave height and $f_p$ is the peak frequency, which is also the reciprocal of the peak period $T_p$.

Parameter $= \begin{cases}\sigma_a, f \le f_p \\ \sigma_b, f > f_p\end{cases}$, for which the adopted values are generally $\sigma_a = 0.07$ and $\sigma_b = 0.09$. Parameter $\gamma$ is known as the

peak enhancement factor, which is related to the bandwidth of Eq. (3). Parameter $B$ in Eq. (3) can also be expressed in terms of $\gamma$ as follows:

$$B = \frac{16\times 0.0624}{0.230+0.0336\gamma-0.185/(1.9+\gamma)}. \tag{4}$$

The adopted directional spreading (Socquet-Juglard et al., 2005) was as follows:

$$D(\theta) = \begin{cases}\frac{2}{\Theta}\cos\left(\frac{\pi\theta}{\Theta}\right)^2 & for \ \ |\theta| \le \Theta/2 \\ 0 & for \ \ |\theta| > \Theta/2\end{cases}, \tag{5}$$

where the spreading parameter $\Theta$ with the unit of degrees (°) or radians (rad) indicates that the energy is distributed within

the range of $\Theta/2$ on both sides of the principal direction (0° or 0 rad). Finally, the 2D initial spectra in the HOS simulations were generated as follows:

$$S(f,\theta) = S(f)D(\theta). \tag{6}$$



The three spectral geometries in Eqs. (3) and (5) are adjustable, and various initial conditions were generated by considering
them in different combinations. Considering Eq. (3), an expression for SP is as follows:

$$\varepsilon = H_s/\lambda_p, \tag{7}$$

where $H_s$ is the significant wave height and $\lambda_p$ is the wavelength at the peak frequency, which can be calculated easily from
$f_p$ according to the linear dispersion relation. Thus, SP was determined by both $H_s$ and $f_p(T_p)$ in Eq. (3). According to
observation in the northern North Sea (deep water) during 1973–2001 (Haver, 2002), $\varepsilon$ is generally <0.06 and most
frequently in the range of 0.01–0.02 (see Fig. 1). To cover all observed sea conditions, $\varepsilon$ of the initial spectra was set to vary
within the range of 0.01–0.06 and, considering the amount of calculation involved in the simulations, the intervals were set
at 0.01. Furthermore, without losing any generality, each value of $\varepsilon$ in the range comprised a unique combination of $H_s$ and
$T_p$, and the settings of SP, together with the corresponding simulated physical and pseudo-spectral space, which are closely
related to $\lambda_p(T_p)$, are listed in Table 1.


The BW and DS of the initial spectra can be determined easily using $\gamma$ and $\Theta$, respectively. For BW, parameter $\gamma$ is set to
vary within the range of 1.0–7.0, in accordance with the JONSWAP observations, with intervals of 1.0. For DS, the range of
the spreading parameter $\Theta$ was set to 8°–340°, and the interval of $\Theta$ in the range of 8°–176° (180°–340°) was set at 8° (20°).

Finally, the total number of initial spectra with adjustable $\varepsilon$, $\gamma$, and $\Theta$ was $6 \times 7 \times 31 = 1302$, and HOSM simulations were
performed on each individually.

## 2.3 Skewness and kurtosis indicators

One complete HOSM simulation with a given initial spectrum is called one realization and the duration of each realization
was $180T_p$. This was because previous experiments revealed that the most significant variations of skewness/kurtosis in the
simulated wave fields all occurred in the first $180T_p$ of the simulation duration, while subsequent changes of both skewness
and kurtosis were minimal because the simulated wave field tended to be Gaussian. The skewness $\mu_3$ and kurtosis $\mu_4$ were
calculated using Eq. (1) based on the $256 \times 256$ surface elevations for each of the output simulated wave fields, and the
output interval was $1T_p$. It should be noted that the time evolutions of $\mu_3$ and $\mu_4$ obtained from one realization were random
and unstable, and that averaging them from a number of realizations with the same initial spectrum would reduce the
uncertainty significantly. In this study, we undertook a series of convergence tests to determine the number of repetitions,
see Sect. 2.4.





We considered indicators of $\mu_3$ and $\mu_4$ that can represent the average characterization of the non-Gaussianity during each $180T_p$ simulation. Therefore, $\mu_3$ and $\mu_4$ time evolutions were first averaged for the final $170T_p$ (denoted as $\overline{\mu_3}$ and $\overline{\mu_4}$),

considering the first $10T_p$ as the relaxation period mentioned in Sect. 2.1. Second, to make the skewness and kurtosis obtained from different initial conditions comparable, we introduced benchmarks denoted as $B_{\mu_3}$ and $B_{\mu_4}$. The benchmarks $B_{\mu_3}$ and $B_{\mu_4}$ were the $\overline{\mu_3}$ and $\overline{\mu_4}$ obtained from the initial condition of $\varepsilon = 0.06$, $\gamma = 10$, and $\Theta = 4°$, representing an extremely steep, near-narrowband, and near-unidirectional sea condition that would hardly ever be observed in a real ocean. Finally, we computed the ratios of the 'averages' to the 'benchmarks' as follows:

$$R_{\mu 3} = \frac{\overline{\mu_4}}{B_{\mu_3}}, R_{\mu 4} = \frac{\overline{\mu_4}}{B_{\mu_4}}, \tag{8}$$

which could be used as non-Gaussianity indicators identifying the skewness/kurtosis time evolution obtained. Thus, the non-Gaussianity of the simulated wave fields corresponding to the initial spectra with certain combinations of $\varepsilon$, $\gamma$, and $\Theta$ were comparable.

**2.4 Number of repetitions**

Incorporation of a large number of repetitions in the averaging process produces results that are more stable and convergent. However, the computational resources available for the entire set of HOSM simulations were rather limited; therefore, eight convergence tests were performed to determine the minimum number of repetitions to be averaged. Each test involved an individual initial condition, as listed in the far-right part of Fig. 2. For each initial condition, 100 realizations were conducted,

of which $n$ were then selected at random to produce a collection named $C_n$. As $n = 30, 40, 50, \dots, 100$, there were eight collections to be gathered. The indicators $R_{\mu 3}$ and $R_{\mu 4}$ of each collection are shown in Fig. 2.

It can be seen from Fig. 2 that satisfactory convergence could be found for both $R_{\mu 3}$ and $R_{\mu 4}$, after 60–70 realizations. Accordingly, the number of repetitions involved in the averaging procedure was set to 80.

**2.5 Spectral geometries and non-Gaussianity indicators**

The HOSM simulations conducted in this study included 1302 initial spectra with various combinations of spectral geometries, as introduced in Sect. 2.2. The simulated results were processed following the procedure outlined above. Thus, the obtained non-Gaussianity indicators could constitute two references for skewness and kurtosis, here denoted as $R_{\mu 3}(\varepsilon, \gamma, \Theta)$ and $R_{\mu 4}(\varepsilon, \gamma, \Theta)$, respectively. Parts of the two references are shown as Fig. 3a–d, where Fig. 3a shows the value

of $R_{\mu 3}$ within the range defined by $\varepsilon$ and $\Theta$ with fixed $\gamma = 3.0$, and similarly, Fig. 3b shows $R_{\mu 3}(\gamma, \Theta)$ with $\varepsilon = 0.02$, Fig. 3c shows $R_{\mu 4}(\gamma, \Theta)$ with $\varepsilon = 0.03$, and Fig. 3d shows $R_{\mu 4}(\varepsilon, \Theta)$ with $\gamma = 3.0$. The $z$-axis in Fig. 3a–d denotes the value of





$R_{\mu3}/R_{\mu4}$, as do the colours in the surface grids, where yellow (blue) denotes a larger (smaller) value. The grids on the surface indicate the discretions of $\varepsilon$, $\gamma$, and $\Theta$ in the initial conditions.

As seen in Fig. 3a–d, the overall trends of $R_{\mu3}$ and $R_{\mu4}$ affected by $\varepsilon$, $\gamma$, and $\Theta$ confirm the statement that the larger the value of $\varepsilon$, the larger the value of $\gamma$, or the smaller the value of $\Theta$, the larger the value of $R_{\mu3}/R_{\mu4}$. Moreover, it can also be seen that the values of $R_{\mu3}$ and $R_{\mu4}$ change continuously with the three geometric parameters. As can be seen from Fig. 3a and 3b, $R_{\mu3}$ is affected most evidently by the SP parameter $\varepsilon$; meanwhile, $\Theta$ can also make $R_{\mu3}$ slightly larger when it is close to 0°, as shown in the upper-left corner of Fig. 3a. Moreover, $\gamma$ can also influence $R_{\mu3}$ within the range where $\Theta$ is narrow, see the

upper-left corner of Fig. 3b. It can be seen in Fig. 3c and 3d that within the range where $\Theta$ is extremely narrow (e.g., $\Theta \leq 20°$), $R_{\mu4}$ decreases markedly as $\Theta$ widens; when $\Theta$ is beyond the extremely narrow range, the value of $R_{\mu4}$ reaches a much lower level and decreases markedly more slowly in comparison with the situation when $\Theta$ widens within the extremely narrow range. The parameters $\gamma$ and $\varepsilon$ can also have certain effects on $R_{\mu4}$. For example, it can be observed that larger values of $\gamma$ and $\varepsilon$ result in a larger value of $R_{\mu4}$, which becomes significant when $\Theta$ is extremely narrow.


As expressed in Eq. (2), kurtosis comprises both a dynamic and a bound part. The dynamic contribution $\mu_4^{free}$ is rather small, i.e., it can be <10% that of the bound component $\mu_4^{bound}$ in a normal sea state (Annenkov and Shrira, 2014), but it can also achieve a very high level in a special wave environment, denoting that MI are active (Fedele, 2015; Janssen, 2003). Otherwise, the value of $\mu_4^{bound}$ generally depends on wave steepness and its change is very limited in comparison with $\mu_4^{free}$

(Janssen, 2009). Thus, the marked change in the value of $R_{\mu4}$ actually quantifies the combinations of $(\varepsilon, \gamma, \Theta)$ that can trigger MI; a more detailed analysis is presented in Sects. 3.3 and 3.4.

It should be noted that $R_{\mu4}$ could take a negative value with some combinations of $(\varepsilon, \gamma, \Theta)$, which could be attributable to two factors. First, some of the simulated values of kurtosis were rather small, and they oscillated slightly and randomly near

the level of 0, resulting in a negative average value. Second, the $\mu_4^{free}$ might exhibit defocusing of the wave energy as the special ratio of BW to DS appeared (Fedele, 2015). Negative $R_{\mu4}$ values represent sea states with less possibility of finding rogue waves; thus, they would not influence identification of MI-triggering combinations.

## 3    Application to rogue wave events

### 3.1    Rogue wave events and wave modelling

The 10 rogue wave events discussed in this section were all observed using laser sensors installed on oil platforms in the North Sea. These include the famous "New Year's wave" captured near the Draupner platform (Cavaleri et al., 2016; Haver, 2004; Janssen, 2015), the rogue wave named Andrea recorded by sensors installed on a bridge between a pair of Ekofisk




Field platforms (Donelan and Magnusson, 2017; Karin Magnusson and Donelan, 2013), and 8 events selected from a continuous record of 421 observations conducted on the North Alwyn platform (Guedes Soares et al., 2003; Slunyaev et al.,

2005; Tomita and Kawamura, 2000). The locations of the platforms and UTC times at which the events occurred are all listed in Table 2, together with the synchronously observed maximum wave heights ($H_{max}$) and significant wave heights ($H_s$), which were digitized from Karin Magnusson and Donelan (2013) for Draupner and Andrea, and from Guedes Soares et al. (2003) for the Alwyn events.

In this study, the wave fields containing the selected events were reproduced using the WWIII wave model ver.5.16 (The WAVEWATCH III R Development Group, 2016). The spectral space modelled was set with 36 directions at intervals of 10° and 35 frequencies spaced from 0.042 Hz up to 1.05 Hz as a geometric progression with the ratio of 1.1. The computational grids of the physical area modelled, illustrated in Fig. 4, comprised an outer grid named NS4 (50°–78°N, −18°E to 15°E) with 0.25° × 0.25° resolution and an inner grid named NS8 (52°–68°N, −6°E to 7°E) with 0.125° × 0.125° resolution. The

bathymetric data were obtained from the ETOPO1 of NGDC (DOI:10.7289/V5c8276m) and the shoreline data were obtained from the GSHHG (Wessel and Smith, 1996). The wind force adopted in the simulations was derived from the ECMWF-ERA5 reanalysis hourly data (DOI: 10.24381/cds.adbb2d47), which provided the $u$–$v$ wind field at 10 m above the sea surface with 0.25° × 0.25° horizontal resolution. The ST4 input ($S_{in}$) and dissipation ($S_{ds}$) source package (Ardhuin et al., 2010) were adopted to work in conjunction with the ERA5 wind force, and the nonlinear wave–wave interactions ($S_{nl}$)

were parameterized using the DIA method (Hasselmann and Hasselmann, 1985b, 1985a). As indicated in the CaseID listed in Table 2, the modelling started from 30 days before and ended in 1 day after the occurrence of Draupner and Andrea events, while for the Alwyn events, the modelling started from 30 days before Alwyn_r1 and ended in 1 day after Alwyn_r8.

Some of the simulated bulk wave parameters are illustrated in Fig. 5a–d together with observed data digitized from

published research for comparison purposes. Comparison of the black solid lines denoting the digitized observations with the blue lines of the simulations reveals acceptable reproduction of the observed sea states, which provides a reliable foundation for the following analyses.

### 3.2 Introducing arbitrary 2D wave spectra into the non-Gaussianity references

As $R_{\mu 3}$ and $R_{\mu 4}$ change continuously with $\varepsilon$, $\gamma$, and $\Theta$, the non-Gaussianity indicators can be obtained easily via 3D

interpolation with an arbitrary combination of $(\varepsilon_i, \gamma_i, \Theta_i)$. The crucial point is to determine how the parameters $\varepsilon_i$, $\gamma_i$, and $\Theta_i$ that appear in the initial spectra might be related to the geometries of SP, BW, and DS in a given spectrum.

The SP for any spectra can be calculated as follows:

$$s_p = \frac{H_{m0}}{L_p}, \tag{9}$$




where $H_{m0} = 4\sqrt{m_0}$ is the significant wave height based on the zeroth-order spectral moment $m_0$, and $L_p$ is the wavelength

at the peak frequency of the given spectrum. Apparently, $s_p$ is always equal to $\varepsilon_i$ with arbitrary spectral shapes.

For DS, the spreading parameter $\sigma_\theta$ (Kuik et al., 1988) can be adopted:

$$\begin{cases} \sigma_\theta = \left[2\left\{1 - \left(\frac{a^2+b^2}{m_0^2}\right)^{1/2}\right\}\right]^{1/2} \\ a = \int_0^{2\pi}\int_0^\infty cos(\theta)\,S(f,\theta)df d\theta \\ b = \int_0^{2\pi}\int_0^\infty sin(\theta)\,S(f,\theta)df d\theta \end{cases}.$$  (10)

By applying Eq. (5) to Eq. (10), it can be found that $\sigma_\theta$ increases monotonically with $\Theta$ within the range of $\Theta \in [8°, 340°]$, as

indicated by the black solid line shown in Fig. 6a. Moreover, it can be fitted by a second-order polynomial, shown as the red

dashed line in Fig. 6a, which can be expressed as follows:

$$\sigma_\theta(\Theta) = -0.0001\Theta^2 + 0.1929\Theta - 0.3576 \ (\Theta \in [8,340]).$$  (11)

Once $\sigma_\theta$ has been calculated from the given spectrum, it is easy to solve the root of Eq. (11) in the range of 8°–340° to

obtain the corresponding $\Theta_i$.

For BW, the parameter $Q_p$ (Goda, 1970) can be used:

$$Q_p = \frac{2}{m_0^2}\int_0^\infty f\left[\int_0^{2\pi} S(f,\theta)d\theta\right]^2 df.$$  (12)

Similar to the above, applying Eq. (3) to Eq. (12) reveals that $Q_p$ increases monotonically with $\gamma$ in the range of $\gamma \in [1,7]$, as

indicated by the black solid line shown in Fig. 6b. It also can be fitted to a second-order polynomial, shown as the red dashed

line in Fig. 6b, which can be expressed as follows:

$$Q_p(\gamma) = -0.0172\gamma^2 + 0.6024\gamma + 1.4946 \ \ (\gamma \in [1,7]).$$  (13)

Thus, parameter $\gamma_i$ could be obtained by solving the root of Eq. (13) in the range of $\gamma \in [1,7]$ with $Q_p$ calculated from the

given spectrum.

**3.3 Non-Gaussianity and spectral geometries in rogue wave sea states**

The parameters $s_p$, $Q_p$, and $\sigma_\theta$ obtained from the modelled wave spectra, together with the related $\varepsilon_i$, $\gamma_i$, and $\Theta_i$ obtained

through the approach introduced above, are shown in Fig. 7 in red and blue colours, respectively. The corresponding non-

Gaussian indicators (here denoted as $R_{\mu3}^{rw}$ and $R_{\mu4}^{rw}$) obtained based on the combination of $(\varepsilon_i, \gamma_i, \Theta_i)$ are shown in Fig. 8. In

both figures, the duration shown for each event extends from 72 hours before to 72 hours after the occurrence of Draupner,

Andrea, or Alwyn_r1. The parameters in Fig. 7 and the non-Gaussianity indicators in Fig. 8 for the events of Draupner,

Andrea, and Alwyn_r1–r8 are depicted by solid, dotted, and dash-dotted lines, respectively. The vertical lines in the two





figures identify the times of rogue wave occurrence, and the values of each parameter at the times are listed in Table 3. The mean values of $R_{\mu3}^{rw}$ and $R_{\mu4}^{rw}$ averaged over the entire modelled duration (approximately one month) are identified by the horizontal dashed lines and are also listed as $\overline{R_{\mu3}^{rw}}$ and $\overline{R_{\mu4}^{rw}}$, respectively, in Table 3.


It can be seen from Fig. 7 and Table 3 that the parameters indicating SP are very similar at the times of occurrence of the selected events, i.e., they are almost all within the range of 0.035–0.040. The parameters indicating BW are slightly different. For example, they are reasonably similar in the Draupner and Andrea events ($Q_p \approx 2.2/\gamma_i \approx 1.3$), whereas decreased values of $Q_p$ and $\gamma_i$ can be found from Alwyn_r1 to Alwyn_r8. For the DS parameters, further differences can be found; however,

the values of $\sigma_\theta$ ($\Theta_i$) always remain above 23° (130°) throughout the duration shown. Moreover, it is obvious that a narrower range of DS values is associated with the occurrences of Draupner and Andrea.

Corresponding to the SP, BW, and DS parameters exhibited in Fig. 7, the non-Gaussianity indicators $R_{\mu3}^{rw}$ and $R_{\mu4}^{rw}$ are shown in Fig. 8. Thanks to the quantitative references proposed in this study, the non-Gaussianity of the sea states containing

the selected rogue waves are comparable. As seen from the upper part of Fig. 8 and Table 3, $R_{\mu3}^{rw}$ at the times of event occurrence have reasonably similar values that are also almost 10% higher than the averaged $\overline{R_{\mu3}^{rw}}$ according to the same benchmark introduced in Sect. 2.3. Thus, it can be concluded that the selected rogue wave events all occurred in non-Gaussian sea states with relatively large skewness. As for $R_{\mu4}^{rw}$, examination of the lower part of Fig. 8 reveals that the kurtosis of the sea states does not seem to present any abnormality in comparison with the averaged $\overline{R_{\mu4}^{rw}}$, and it can be found

in Table 3 that both $R_{\mu4}^{rw}$ and $\overline{R_{\mu4}^{rw}}$ have reasonably low values according to the current benchmark. Moreover, the values of $R_{\mu4}^{rw}$ in the Draupner and Andrea events are even lower than the corresponding $\overline{R_{\mu4}^{rw}}$, even though more unidirectional wave environments that might be more conducive to triggering MI were found.

As skewness is contributed entirely by second-order nonlinearities, and given the higher-value $R_{\mu3}^{rw}$ and lower-value $R_{\mu4}^{rw}$, it

is inferred that the selected sea states were dominated by second-order nonlinearities and that third-order MI were suppressed in these selected events. To confirm these inferences, additional HOSM simulations were undertaken in this study. The additional simulations were performed based on the same experimental environment established in Sect. 2.1. However, the initial conditions were replaced by the modelled wave spectra of the 10 events, and simulations that considered the second-order nonlinearities (M = 2) only and both the second- and the third-order nonlinearities (M = 3) were undertaken

separately to elucidate the dominant mode in those wave fields. A similar approach can be found in previous research (Fedele et al., 2016), in which the inactive MI in the Draupner and Andrea events were also studied.




The results of the additional simulations are illustrated in Fig. 9. As shown in the upper-right corner of Fig. 9, each panel titled with a CaseID contains an upper and lower part exhibiting the skewness and kurtosis of the simulated wave field, respectively, and the *x*-axis in each panel represents the time duration with a dimensionless form of $Time/T_p$. The black and red lines shown in both parts depict the skewness/kurtosis time evolution simulated when considering M = 2 and M = 3, respectively. Certainly, no discrepancies exist between the black and red lines in the upper part of each panel because skewness is contributed entirely by the second-order nonlinearities. However, the lack of significant discrepancies between the two sets of lines in the lower part of each panel indicates that kurtosis is also dominated solely by the second-order nonlinearities. Therefore, it is confirmed that second-order nonlinearities were dominant in the rogue wave sea states and that the third-order MI were inactive in the selected events.

Moreover, the blue dashed lines in each panel of Fig. 9 denote the indicators of $R_{\mu3}^{rw}$ and $R_{\mu4}^{rw}$, which have been respectively multiplied by the benchmarks of $B_{\mu_3}$ and $B_{\mu_4}$ (see Sect. 2.3) to be comparable with the simulated black and red lines. Acceptable goodness of the fit of the blue dashed lines to the simulated black and red lines proves the feasibility of the newly developed approach of applying the proposed non-Gaussianity references to real wave spectra.

### 3.4 Why MI are inactive in wind-dominated sea states

In this section, from the perspective of spectral geometries, we elucidate why MI were inactive in the selected rogue wave events. First, the 10 events all occurred in stormy sea states, which had been fully dominated by wind-sea systems before the events occurred. This can be proven by introduction of a 'wind-sea fraction' (Hanson and Phillips, 2001; Tracy et al., 2007):

$$W = E^{-1}E_{U_p>c}, \tag{14}$$

where $E$ is the total spectral energy, and $E_{U_p>c}$ is the energy in the spectrum for which the projected wind speed $U_p$ is larger than the local wave phase velocity $c$. Parameter $U_p$ can be expressed as follows:

$$U_p = C_{mult}U_{10}\cos(\delta), \tag{15}$$

where $U_{10}$ is the wind speed at 10 m above the sea surface, $\delta$ is the angle between the direction of wave propagation and the direction in which the wind is blowing, and $C_{mult}$ is a multiplier with a value of 1.7. The wind-sea fraction in each of the selected events is illustrated in Fig. 10, where both the durations exhibited and the styles of depiction of the events are the same as in Figs. 7 and 8. It can be seen in Fig. 10 that the wind-sea fraction accounts for nearly all the spectral energy in these selected events, except for the final three Alwyn events; however, even for the three exceptions, the fraction still accounts for >80% of the total spectral energy. Absolute dominance of the wind-sea state began at least 20, 10, and 30 hours before the occurrence of Draupner, Andrea, and Alwyn_r1, respectively, indicating that the wind waves had grown sufficiently when those events occurred.



For dominant wind-sea wave fields, it is known that the spectral geometry of DS is generally wide, although the range of DS
might become narrower as the waves become more developed. It can be seen from Fig. 11a how the indicator $R_{\mu_4}$ varies
with the geometry $\Theta$ when the other two geometries are fixed as $\varepsilon = 0.06$, $\gamma = 7.0$ (black solid line), $\varepsilon = 0.06$, $\gamma = 2.0$
(blue dashed line), $\varepsilon = 0.04$, $\gamma = 7.0$ (red dashed line), and $\varepsilon = 0.04$, $\gamma = 2.0$ (black dashed line), and the pentagrams
denote the values of $R_{\mu 4}^{rw}$ with the corresponding $\Theta_i$ at the occurrences of the selected events. As expected, in Fig. 11a, the
value of $R_{\mu_4}$ decreases dramatically as $\Theta$ increases, and the pentagrams are obviously located outside the region in which $\Theta$
is sufficiently narrow to result in large $R_{\mu_4}$, even though the $\Theta_i$ was slightly narrower in the Draupner and Andrea events.
With $\Theta = \Theta_i > 130°$, the values of $R_{\mu_4}$ would not be raised significantly, even though $\varepsilon$ becomes steeper (blue dashed line
in Fig. 11a) or $\gamma$ becomes narrower (red dashed line), or both parameters take larger values (black solid line). Therefore, it
can be concluded that the small values of $R_{\mu 4}^{rw}$ obtained in the selected events are mainly due to the broad DS in the wind-sea
wave fields.


It is also known that, as the wind-sea wave field develops, BW becomes broader, and this phenomenon can be observed in
the Alwyn events. Similarly, Fig. 11b shows how indicator $R_{\mu_4}$ varies with geometry $\gamma$ when the other two geometries are
fixed as $\varepsilon = 0.06, \Theta = 8°$ (black solid line), $\varepsilon = 0.06, \Theta = 128°$ (blue dashed line), $\varepsilon = 0.04, \Theta = 8°$ (red dashed line), and
$\varepsilon = 0.04, \Theta = 128°$ (black dashed line), and the pentagrams denote the values of $R_{\mu 4}^{rw}$ with the corresponding $\gamma_i$. Comparing
the black solid/red dashed line with the blue dashed/black dashed line in Fig. 11b, it can be seen that with an extremely
narrow DS ($\Theta = 8°$), the values of $R_{\mu_4}$ can be raised significantly. However, with $\gamma = \gamma_i < 2.0$ obtained in the selected
events, the value of $R_{\mu_4}$ remains low, although it can be raised significantly as parameter $\gamma$ becomes larger, especially when
the range of DS is extremely narrow, as shown by the black solid and red dashed lines in Fig. 11b. Thus, it can be concluded
that the broad BW representing developed wind waves might also suppress the value of $R_{\mu_4}$, which might result in inactive
MI in such sea states.

As for the skewness indicator, as described in Sect. 2.5, $R_{\mu_3}$ is affected most evidently by geometry parameter $\varepsilon$. The
parameter $\Theta = \Theta_i > 130°$ and $\gamma = \gamma_i < 2.0$ obtained in the rogue wave sea states would not have significant influence on
the indicator. Therefore, the higher values of $R_{\mu 3}^{rw}$ at the times of rogue wave occurrence are mainly due to the higher $s_p/\varepsilon_i$.
According to Fig. 1, values of $\varepsilon_i$ within the range of 0.035–0.04, as mentioned in Table 3, are relatively large and rare in
field observations.

## 4   Conclusions and discussion

In this study, we established a set of references that allowed quantitative comparison of the non-Gaussianity of sea states
with various combinations of three spectral geometries: SP, BW, and DS. The non-Gaussianity references were established





based on numerous HOSM simulations, in which various 2D wave spectra were adopted as initial conditions and non-Gaussianity indicators were obtained from the simulated non-Gaussian sea states. The connections between the SP, BW, and DS of the initial spectra and the corresponding non-Gaussianity indicators obtained then constituted the references. We also applied the references to some real rogue wave events. The rogue wave sea states were reproduced using the spectral wave model WWIII, and an approach to introduce the modelled 2D spectra into the references was developed. Then, analyses

focusing on the three spectral geometries and their corresponding non-Gaussianity in the selected sea states were performed.

It was found that all the selected rogue waves occurred in wave fields that were entirely dominated by wind-sea systems, and that waves had become well developed before those events occurred. From the perspective of non-Gaussianity, the sea states all presented greater skewness and less kurtosis when the events occurred, indicating that the selected sea states were

dominated by second-order nonlinearities and that the third-order MI were suppressed in such sea states (as confirmed by additional HOSM simulations). From the perspective of spectral geometries, SP was relatively steep when those events occurred, whereas BW and DS were both reasonably broad. According to the references, it was the steeper SP that was closely related to the greater non-Gaussianity of skewness, and it was the broader BW and DS that might have suppressed the third-order MI, resulting in less kurtosis in those selected sea states.


It is known that wind waves propagate multi-directionally, while active MI can only be observed in unidirectional wave fields. Therefore, it can be inferred qualitatively that rogue waves occurring in wind-sea states with large DS cannot be formed by the MI. However, narrower DS might be observed at the precise time of occurrence of some rogue events, e.g., the Draupner and Andrea events investigated in this study. The newly proposed non-Gaussianity references provided

quantitative support regarding this topic, and it was confirmed that the DS in the selected events was far from the range in which MI could be triggered, even with the narrower DS observed in the Draupner and Andrea events. Furthermore, still based on the quantitative references, it was found that the broad BW observed in developed waves might also have created an unsuitable environment for the generation of MI. In fact, rogue waves with extremely large wave height and extreme destructive power, such as the Draupner and Andrea waves, generally occur in stormy sea states, where the well-developed

wave environment could provide energy to support such rogue events. However, owing to the relatively broad BW and DS in such sea states, such giant rogue waves cannot be formed by the third-order quasi-resonant nonlinearities and the associated instabilities.

In addition to providing quantitative support, the newly proposed references, which were established based on the HOSM

simulations, allow arbitrary spectral width to be involved without the narrowband or unidirectional limitations. Moreover, in conjunction with the approach developed to introduce arbitrary 2D spectra into the references, the applicability of the references to investigations of real rogue wave sea states was demonstrated.

With respect to the real ocean, our model is surely an oversimplification. For example, it focuses purely on the
nonlinearities between waves, ignoring other physical mechanisms that might influence the non-Gaussianity.
Furthermore, it was established based on only unimodal spectral shapes, ignoring bi-modal and even multi-
modal shapes, even though such shapes might result in wide BW and DS that could make the sea state less
conducive to rogue wave occurrence. Nevertheless, this study provided a new perspective for the study of rogue
wave sea conditions, and further research could be undertaken on this basis.

**Author Contribution**

The paper and its methodology were conceptualized and developed by Jiang Xingjie, who also conducted the experiments
and data analysis work. The original draft was wrote by Jiang Xingjie, and other co-authors also contributed to preparing,
editing, drawing, etc.

**Competing interests**

The authors declare that they have no conflict of interest.

**Acknowledgements**

This work was supported by the National Key Research and Development Program of China (Nos. 2016YFC1401805,
2016YFC1402004). We thank Guillaume Ducrozet and Yves Perignon from the LHEEA of the École Centrale de Nantes
and CNRS for their great assistance in helping us understand the HOS method and the use of HOS-ocean. We thank James
Buxton MSc from Liwen Bianji, Edanz Group China (www.liwenbianji.cn/ac), for editing the English text of this manuscript.

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


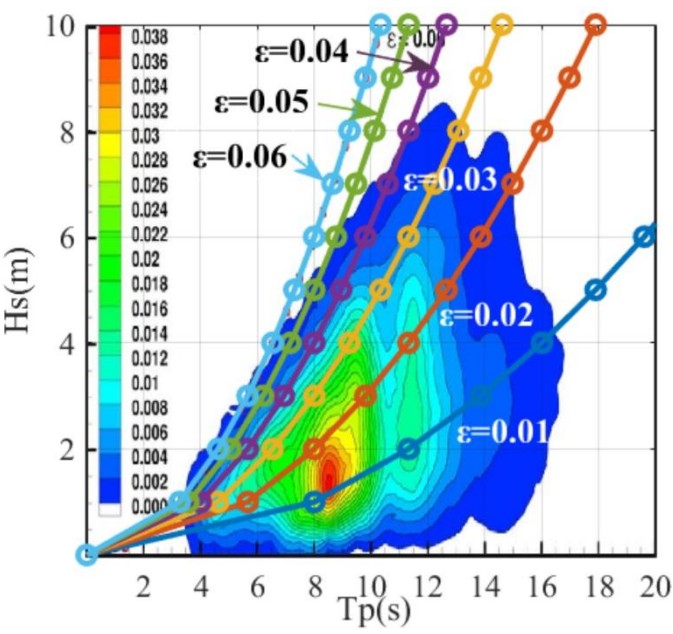

**Figure 1. Combined $H_s$–$T_p$ distribution observed in the northern North Sea during 1973–2001.**
**(The figure is taken from Fig. 2 of Ducrozet et al. (2017) and only the line of $\varepsilon = 0.06$ is original; the lines denoting $\varepsilon = 0.01 - 0.05$ were added by the authors.)**


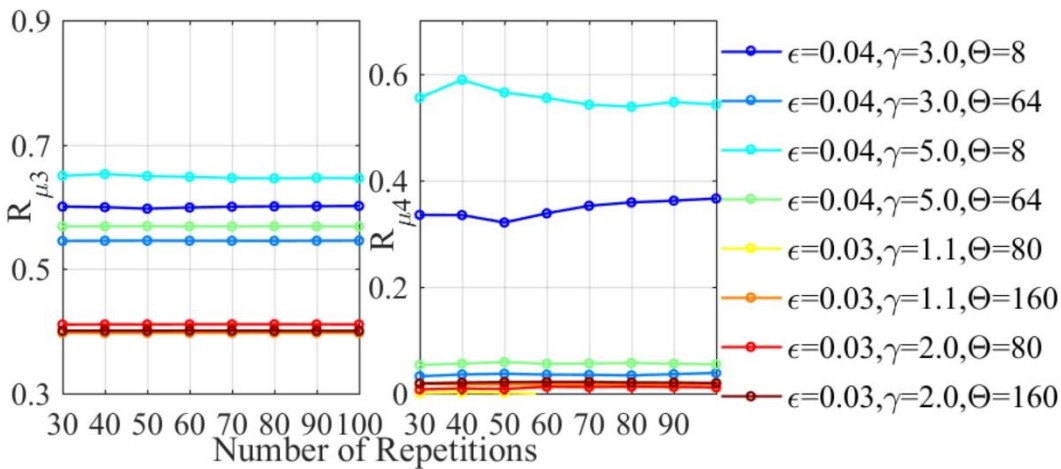

**Figure 2. Results of convergence tests for the number of repetitions participating in the averaging process.**



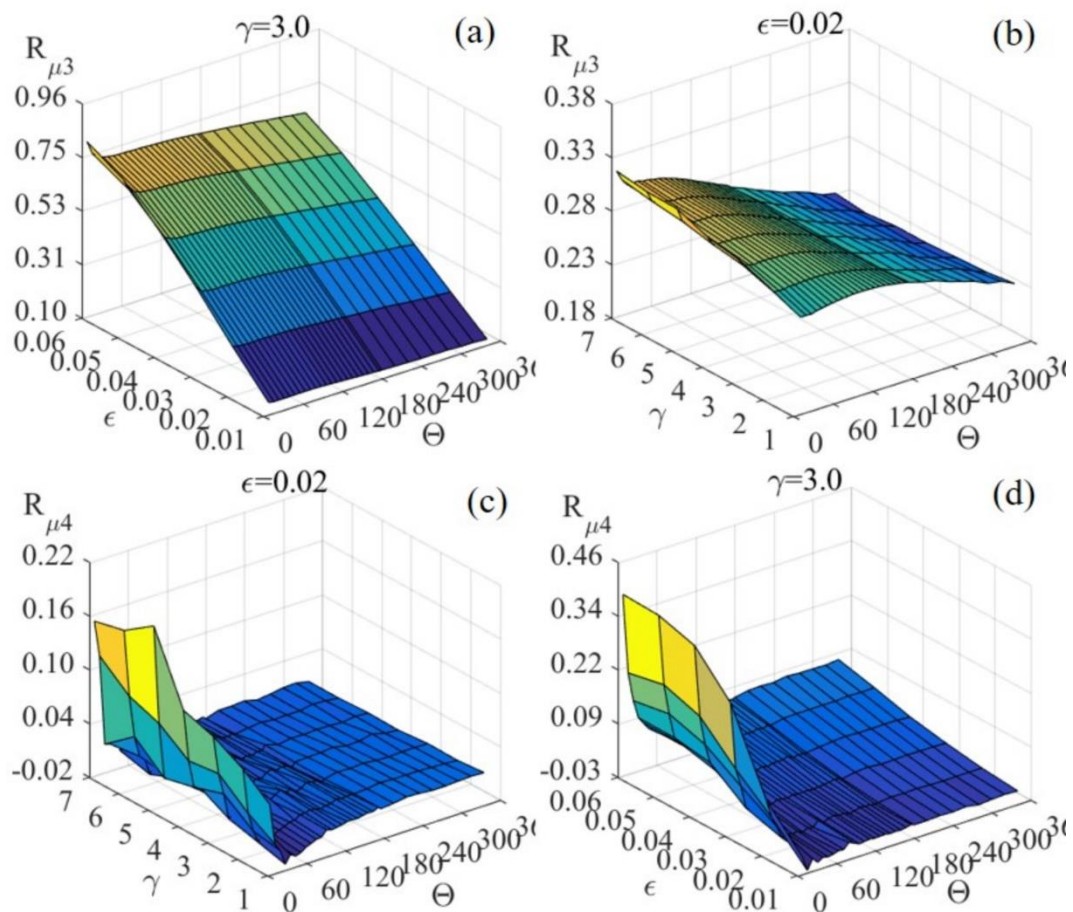

**Figure 3. Parts of the non-Gaussianity references:**
for $R_{\mu 3}$ with (a) $\gamma = 3.0$ and (b) $\varepsilon = 0.02$, and for $R_{\mu 4}$ with (c) $\varepsilon = 0.02$ and (d) $\gamma = 3.0$.


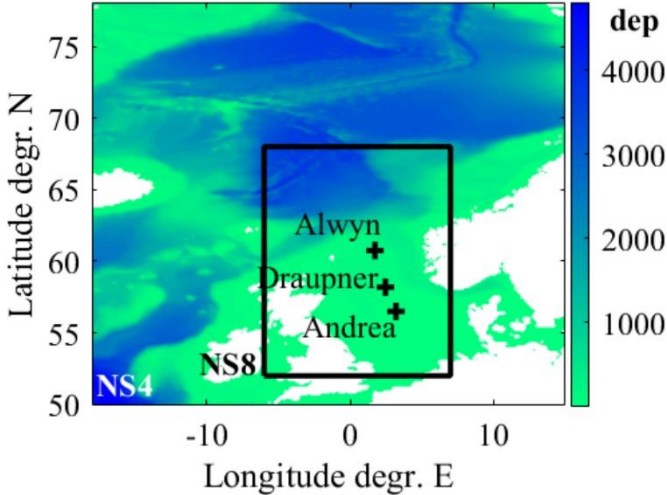

**Figure 4. Computational grids of the North Sea.**

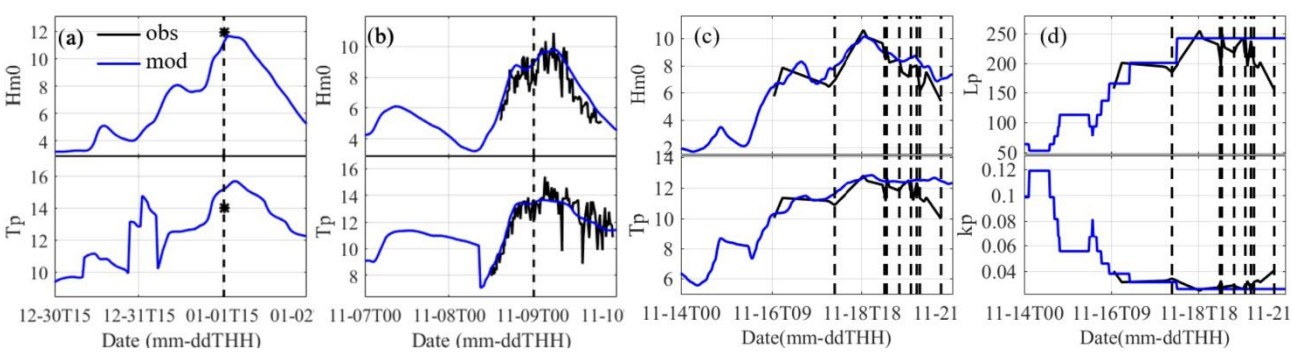

**Figure 5. Comparison of simulated results and observations for (a) Draupner, (b) Andrea, and both (c) and (d) Alwyn.**
**(The blue lines depict simulated results and the black lines (asterisk) depict the digitized observations from Table 1 and Fig. 3 of**
**Magnusson and Donelan (2013) for (a) & (b), and from Table 1 of Guedes Soares et al. (2013) for (c) & (d). The vertical dashed**
**lines identify the times of rogue wave occurrence.)**

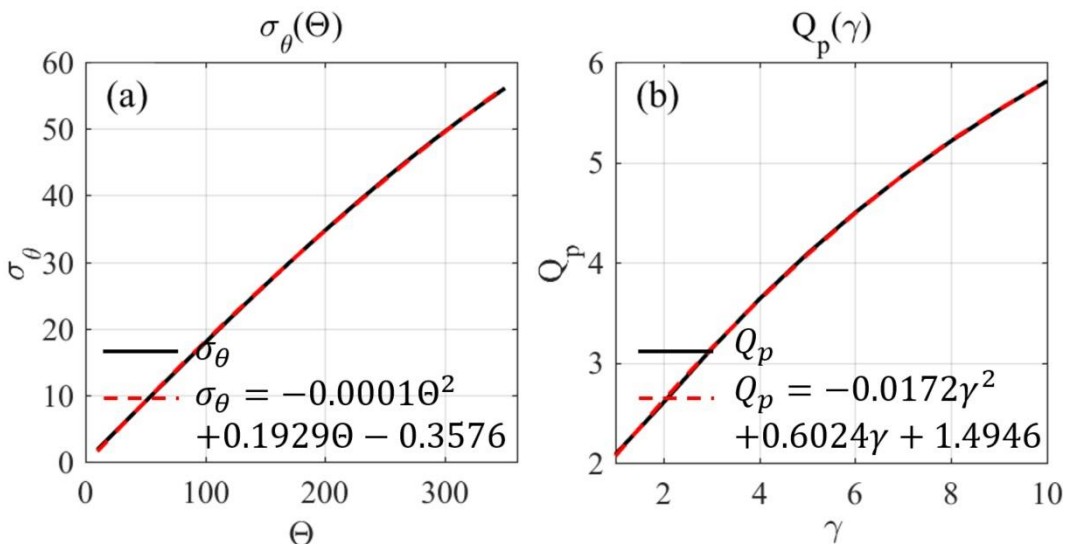

**Figure 6. Polynomials fitted for (a) $\sigma_\theta(\Theta)$ and (b) $Q_p(\gamma)$.**


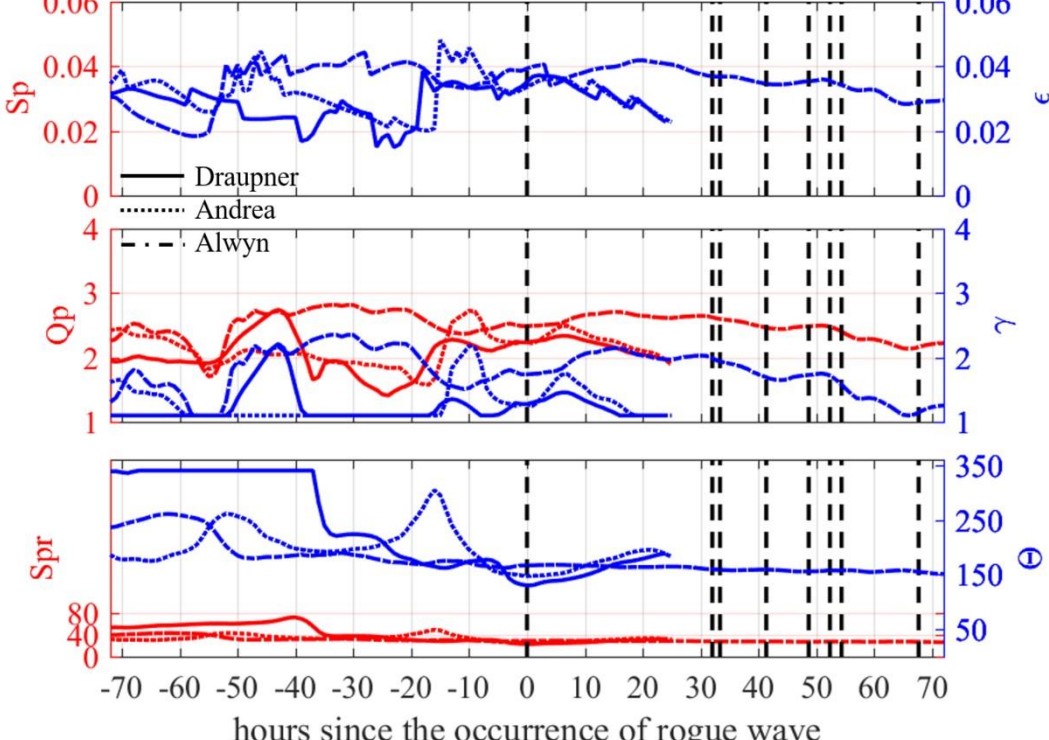

**Figure 7. Spectral geometries during the rogue wave events.**
**(Parameters in Draupner, Andrea, and Alwyn events are depicted by solid, dotted, and dash-dotted lines, respectively. The vertical dashed lines identify the times of rogue wave occurrence.)**






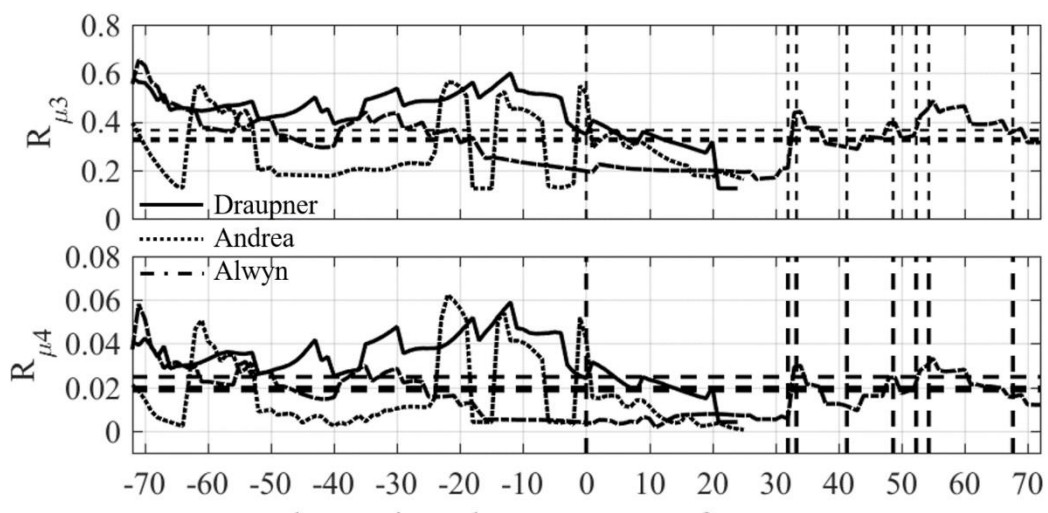

**Figure 8. Non-Gaussianity indicators during the rogue wave events.**
(Parameters in Draupner, Andrea, and Alwyn events are depicted by solid, dotted, and dash-dotted lines, respectively. The vertical dashed lines identify the times of rogue wave occurrence, and the horizontal lines identify the mean values of $R_{\mu 3}^{rw}/R_{\mu 3}^{rw}$ within the modelled duration.)


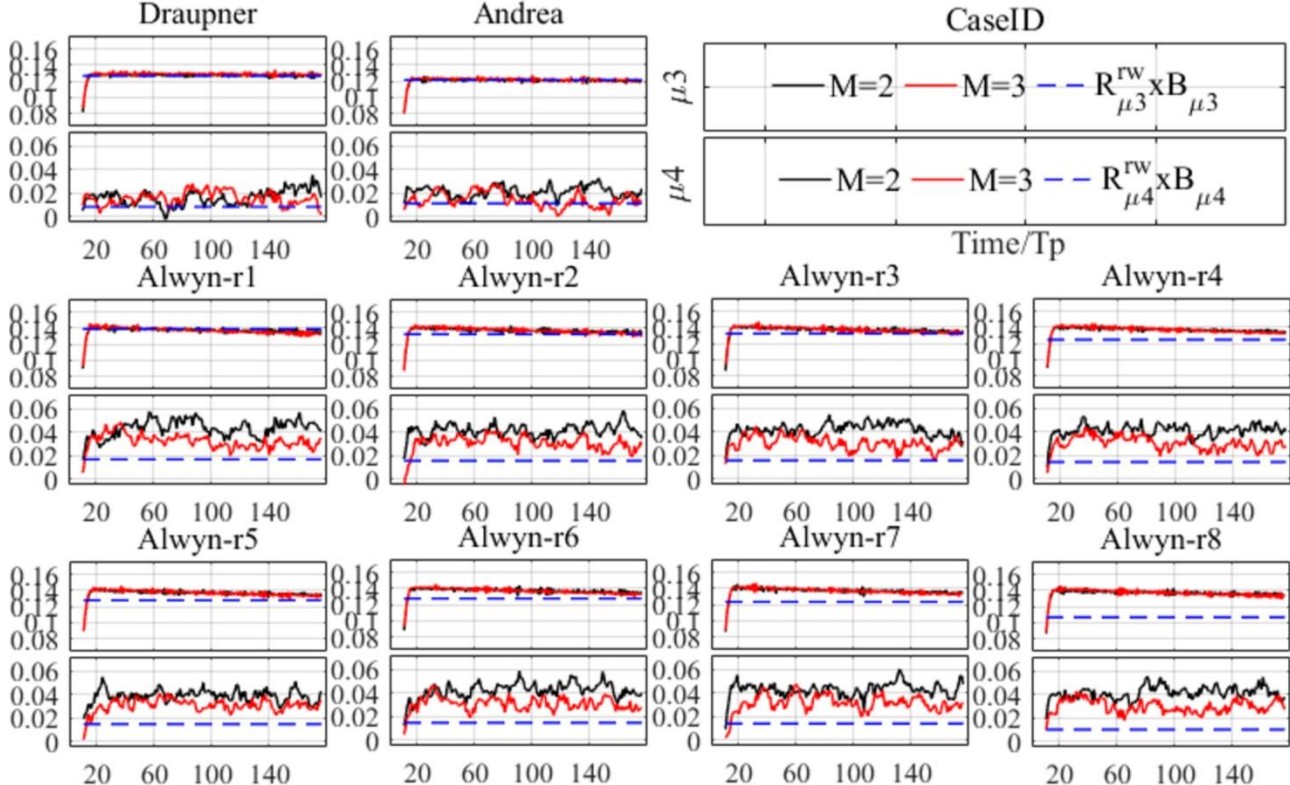

**Figure 9. Results of the additional HOSM simulations.**

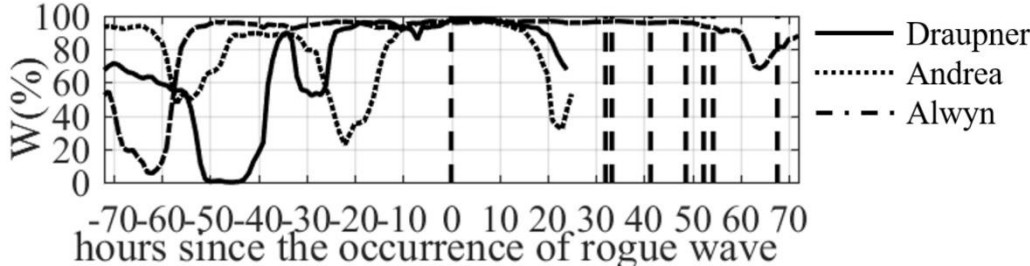

**Figure 10. Wind-sea fraction (W) in the selected rogue wave sea states.**
(The wind-sea fractions in the Draupner, Andrea, and Alwyn events are depicted by solid, dotted, and dash-dotted lines respectively. The vertical dashed lines identify the times of rogue wave occurrence.)

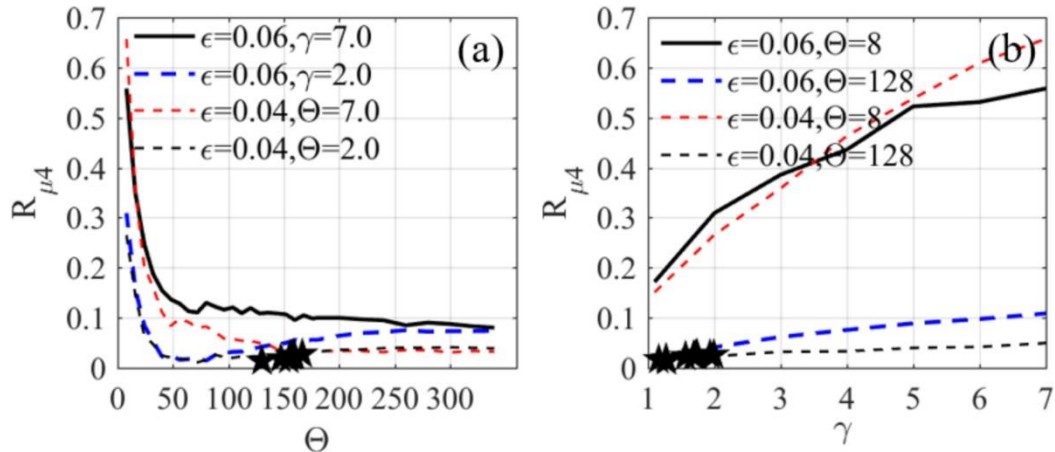

**Figure 11. Variation of $R_{\mu_4}$ with (a) $\Theta$ and (b) $\gamma$ when the other geometries are fixed.**


**Table 1. Settings of SP and the corresponding simulated physical and pseudo-spectral space**

| $\varepsilon$ | $H_s(m)$ | $T_p(s)$ | $f_p(Hz)$ | $\lambda_p(m)$ | $k_p$ | $k_{max}$ | $f_{max}(Hz)$ | $L_x(L_y)(m)$ | $\Delta k$ |
|---|---|---|---|---|---|---|---|---|---|
| 0.01 | 1.0 | 8.0 | 0.1250 | 100.00 | 0.0628 | 0.3142 | 0.28 | 2550.0 | 0.0025 |
| 0.02 | 3.0 | 9.8 | 0.1020 | 150.00 | 0.0419 | 0.2094 | 0.23 | 3825.0 | 0.0016 |
| 0.03 | 6.0 | 11.3 | 0.0884 | 200.00 | 0.0314 | 0.1571 | 0.20 | 5100.0 | 0.0012 |
| 0.04 | 7.0 | 10.6 | 0.0945 | 175.00 | 0.0359 | 0.1795 | 0.21 | 4462.5 | 0.0014 |
| 0.05 | 4.0 | 7.2 | 0.1397 | 80.00 | 0.0785 | 0.3927 | 0.31 | 2040.0 | 0.0031 |
| 0.06 | 2.0 | 4.6 | 0.2164 | 33.33 | 0.1885 | 0.9426 | 0.48 | 849.9 | 0.0074 |

**Table 2. Basic information of the studied rogue wave events.**

| CaseID | Time(UTC) | Latitude(N) | Longitude(E) | $H_{max}(m)$ | $H_s(m)$ |
|---|---|---|---|---|---|
| Draupner | 1995.01.01T15:20:00 | 58°11′19.30″ | 2°28′0.00″ | 25.0 | 11.9 |
| Andrea | 2007.11.09T00:54:22 | 56°30′0.00″ | 3°12′0.00″ | 21.1 | 9.2 |
| Alwyn_r1 | 1997.11.18T01:10:00 | 60°45′0.00″ | 1°44′0.00″ | 16.4 | 6.9 |
| Alwyn_r2 | 1997.11.19T09:11:00 | 60°45′0.00″ | 1°44′0.00″ | 18.0 | 8.9 |
| Alwyn_r3 | 1997.11.19T10:31:00 | 60°45′0.00″ | 1°44′0.00″ | 20.3 | 9.1 |
| Alwyn_r4 | 1997.11.19T18:31:00 | 60°45′0.00″ | 1°44′0.00″ | 18.1 | 8.5 |
| Alwyn_r5 | 1997.11.20T01:51:00 | 60°45′0.00″ | 1°44′0.00″ | 18.2 | 7.9 |
| Alwyn_r6 | 1997.11.20T05:31:00 | 60°45′0.00″ | 1°44′0.00″ | 17.0 | 8.0 |
| Alwyn_r7 | 1997.11.20T07:31:00 | 60°45′0.00″ | 1°44′0.00″ | 13.5 | 6.1 |
| Alwyn_r8 | 1997.11.20T20:51:00 | 60°45′0.00″ | 1°44′0.00″ | 11.7 | 5.4 |


**Table 3. Spectral geometries and non-Gaussian indicators at times of rogue wave occurrence**

| CaseID | $S_p/\varepsilon_i$ | $Q_p$ | $\sigma_\theta$ | $\gamma_i$ | $\Theta_i$ | $R_{\mu_3}^{rw}$ | $R_{\mu_4}^{rw}$ | $\overline{R_{\mu_3}^{rw}}$ | $\overline{R_{\mu_4}^{rw}}$ |
|---|---|---|---|---|---|---|---|---|---|
| Draupner | 0.035 | 2.24 | 23.1 | 1.28 | 130.2 | 0.460 | 0.014 | 0.3648 | 0.0248 |
| Andrea | 0.034 | 2.23 | 25.9 | 1.27 | 147.3 | 0.442 | 0.018 | 0.3205 | 0.0199 |
| Alwyn_r1 | 0.039 | 2.49 | 29.0 | 1.74 | 166.7 | 0.505 | 0.028 | 0.3301 | 0.0185 |
| Alwyn_r2 | 0.037 | 2.63 | 27.9 | 1.99 | 159.5 | 0.482 | 0.026 | 0.3301 | 0.0185 |
| Alwyn_r3 | 0.037 | 2.60 | 27.7 | 1.94 | 158.7 | 0.481 | 0.026 | 0.3301 | 0.0185 |
| Alwyn_r4 | 0.035 | 2.46 | 27.8 | 1.69 | 159.1 | 0.453 | 0.024 | 0.3301 | 0.0185 |
| Alwyn_r5 | 0.035 | 2.49 | 27.3 | 1.73 | 155.9 | 0.464 | 0.024 | 0.3301 | 0.0185 |
| Alwyn_r6 | 0.035 | 2.48 | 27.5 | 1.71 | 157.5 | 0.463 | 0.024 | 0.3301 | 0.0185 |
| Alwyn_r7 | 0.034 | 2.40 | 27.6 | 1.57 | 157.6 | 0.449 | 0.023 | 0.3301 | 0.0185 |
| Alwyn_r8 | 0.029 | 2.17 | 27.1 | 1.16 | 154.5 | 0.387 | 0.016 | 0.3301 | 0.0185 |