# Peer review of "Numerical investigation on spectral geometries and their relation to non-Gaussianity in sea states with occurrence of rogue waves: windsea dominated events"

_Natural Hazards and Earth System Sciences, 2020_

## Referee Comment (RC1) · Anonymous Referee #1 · 9 Nov 2020

I cannot support this manuscript for publication in the present form. Here is comments and major concern of this manuscript.

1. General comments 1.1. Scientific significance I think that this manuscript should clarify its novelty in comparison with previous studies.

This manuscript investigated the relationship between the non-Gaussianity of the sea states and the three spectral parameters: SP, BW, and DS. On the other hand, some previous studies showed theories to predict kurtosis as the indicator of the non-

Gaussianity from spectral parameters, as mentioned in line 51-53 of this manuscript. In line 53-55, the authors say that "it remains difficult to assess directly the severity of the deviation from Gaussianity for a given sea state, or to infer the nature of the dominant nonlinearities." I didn't figure out why the authors argued "it remains difficult". I believe that the existing theories mentioned in line 51-53 can be applied to predict the non-Gaussianity for operational wave forecasts. Therefore, I cannot understand why HOSM was required to predict the kurtosis for the rogue wave events mentioned in the manuscript. Since the line 53-55 corresponds to the key issue of this manuscript, I think these lines should be written in more detail.

Additional comments on the novelty are the following: ïĄňEven if HOSM is necessary, Xiao et al. (2013) conducted similar computation of HOSM to relate kurtosis to spectral parameters. Is not the result of Xiao et al. (2013) insufficient? ïĄňIn order to clarify the novelty and deepen the discussion, I recommend the authors to compare their results in Figure 9 with the existing theories. ïĄňThe conclusion of this manuscript is fairly similar to that of Fedele et al. (2016).

1.2. Scientific quality

There are a lack of evidence and a leap in logic in this manuscript. Please see 2. Specific comments.

1.3. Presentation quality

This manuscript is well organized. The figures and tables are easy to see. I found no problem regarding usage of English language. 2. Specific comments

Line 58-60: With regard to "the magnitude of the kurtosis or the narrowness of the DS required for triggering the MI remains unclear. Moreover, the question of whether there are any other thresholds for the SP and BW geometries remains to be resolved", Ribal et al. (2013) derived criterion of the modulational instability for JONSWAP spectra.

Line 71: The authors should explain the meaning of "involve additional complicated

factors might influence estimation of the non-Gaussianity ..." more clearly. What are "additional complicated factors"?

Line 359-360: With regard to "For dominant wind-sea wave fields, it is known that the spectral geometry of DS is generally wide, although the range of DS might become narrower as the waves become more developed", please cite some references or show some evidences.

Line 338-341: The authors argued that predictions of the kurtosis based on the spectral parameters (the blue line in Figure 9) are comparable to kurtoses of additional HOSM simulation based on the modeled spectra of the WAVEWATCHIII (the red line in Figure 9). However, the latter is several times larger than the former in all cases of Alwyn, actually. Some explanation on this discrepancy are needed.

Line 125, Table 2 and Table 3: Please add information about water depth of Draupner, Andrea, and Alwyn site because the water depth strongly affects the modulational instability (Janssen & Onorato, 2007). The infinite water depth was adopted for HOSM simulation (line 125) in this study, but was this assumption valid for these sites?

3. Technical comments

Title: The title is long.

In "Abstract" and "Conclusion and Discussion": Two different fonts are mixed in these sections.

Line 234, 263, and 290: Line spacings of headings of sections 3.1, 3.2, and 3.3 are different from that of previous sections. The former is narrower than the latter.

References Fedele, F., Brennan, J., Ponce de León, S., Dudley, J. M., Dias, F., De León, S. P., Dudley, J. M., & Dias, F. (2016). Real world ocean rogue waves explained without the modulational instability. Scientific Reports, 6(27715), 1–11. https://doi.org/10.1038/srep27715 Janssen, P. A. E. M., & Onorato, M. (2007). The Intermediate Water Depth Limit of the Zakharov Equation and Consequences for Wave Prediction. Journal of Physical Oceanography, 37(10), 2389–2400. https://doi.org/10.1175/JPO3128.1 Ribal, A., Babanin, A. V, Young, I. R., Toffoli, A., & Stiassnie, M. (2013). Recurrent solutions of the Alber equation initialized by Joint North Sea Wave Project spectra. Journal of Fluid Mechanics, 719, 314–344. https://doi.org/10.1017/jfm.2013.7 Xiao, W., Liu, Y., Wu, G., & Yue, D. K. P. (2013). Rogue wave occurrence and dynamics by direct simulations of nonlinear wave-field evolution. Journal of Fluid Mechanics, 720, 357–392. https://doi.org/10.1017/jfm.2013.37

---

## Author Comment (AC1) · 20 Nov 2020

Reply to the comments from anonymous referee#1:

We sincerely thank referee#1 for the valuable feedback that we have used to improve the quality of our manuscript. The referee's comments are laid out in *italicized font* and the comments have been numbered in the authors' response. Our response is given in normal font and changes/additions to the manuscript are given in blue text. The authors' response can be found in the attachment.

Sincerely Yours,
Dr. Jiang Xingjie

Authors' Response:

This manuscript aims at providing a systematic approach that allows a convenient and quantitative comparison of non-Gaussianity of real-world wave fields through the corresponding wave spectra. The newly proposed approach includes: i) a set of pre-calculated references representing the relation of non-Gaussianity to spectral geometries, and ii) an approach to introduce arbitrary 2D spectra into the references. Since the occurrence of rogue waves is closely related to the two issues: spectral geometries and non-Gaussianity of sea states, we applied this approach to some rogue events occurred in real oceans. The results confirmed with the existing theories and conclusions, and provided a quantitative support regarding the topic of "explaining formation of rogue waves without modulational instabilities" in wind-sea dominated sea states. Apparently, the newly proposed approach is operational and can be used in more studies related to rogue wave sea states.

Comment 1: *This manuscript should clarify its novelty in comparison with previous studies.* And *"Line 58-60: With regard to …, Ribal et al. (2013) derived criterion of the modulational instability for JONSWAP spectra", "Even if HOSM is necessary,…. Is not the result of Xiao et al. (2013) insufficient?",* and *"The conclusion of this manuscript is fairly similar to that of Fedele et al. (2016)".*
Response: We must admit that the statement "it remains difficult …" in the original manuscript was a bit ambiguous and the following example was not proper. As mentioned in the referee's comments, there are existed theories and approaches to obtain the non-Gaussianity indicators through the three geometries, and some of the approaches can even be applied to operational rogue wave forecast systems. However, these 'operational' indicators are not suitable to assess the severity of the deviation from Gaussianity for a real sea state, this is because:

1) The operational non-Gaussianity indicators, especially dynamic kurtosis, were first obtained from theoretical models derived under the narrowband assumption in an environment with near-unidirectional wave propagation. The original expression of those indicators cannot be applied to the real sea conditions with broad bandwidth and directional spreading.

2) In the rogue wave forecast systems, specific parameters representing the geometries were selected and undermined parameters were introduced to calibrate the indicators to adapt the real wave environment. However, calibrations were conducted based on the final forecast results, i.e., the exceptional maximum wave/crest height, rather than the skewness/kurtosis observed in real environment. The skewness/kurtosis of wave surface depends on the number of waves involved

in the statistics, and the scope of the statistical sea surface and the length of the statistical time can hardly be defined. Therefore, the number of waves is a quantity that cannot be accurately estimated in real wave fields. Furthermore, the number of waves is also an important factor determines the exceptional wave/crest heights forecasted, together with the skewness/kurtosis indicators in the forecast systems. (And that is the "additional complicated factors").

Details of introducing and calibrating the operational non-Gaussianity indicators can be found in Barbariol et al. (2015; 2017) for the WWIII-STE system, and in ECMWF Technical Memorandums (Janssen, 2017; Janssen and Bidlot, 2009) for the ECMWF-IFS system. (Barbariol et al., 2015, 2017; Janssen, 2017; Janssen and Bidlot, 2009)

In this study, we used HOSM to simulate the wave field. The HOSM can adopt arbitrary 2D wave spectra as initial conditions without any limitation of spectral shape. To avoid the impact of the number of waves on the statistics of skewness/kurtosis, the number of waves was set to be almost the same (25.5*25.5) in the initial field of each HOSM simulation (see Sect. 2.1), and the ratios denoted as $R_{\mu3}$ and $R_{\mu4}$ were introduced (see Eq. (8) in Sect. 2.3) to represent the non-Gaussianity obtained under these similar experimental environments.

As mentioned in the comment, Fedele et al. (2016) produced similar results. In fact, HOSM could be very cumbersome if it is applied to every spectrum obtained in the time duration of interest. As the HOSM simulations were carried out 50 times for each spectrum, Fedele's work only focused on three spectra which represented the sea states where and when the events of Draupner, Andrea and Killard occurred. In this study, thanks to the pre-calculated references and the introducing approach, non-Gaussianity indicators $R_{\mu3}$ and $R_{\mu4}$ could be obtained very conveniently. Then, the evolution of non-Gaussianity within concerned wave fields and time durations can be presented, as exhibited in Fig. 8. Non-Gaussianity evolution shown in Fig. 8 is a unique production of this study, through Fig. 9 exhibited similar results as Fedele's work. (Fedele et al., 2016)

Previous research like Ribal et al. (2013) (Ribal et al., 2013) and Xiao et al. (2013) (Xiao et al., 2013) have provided criterions containing the three geometries for MI. And these criterions were derived based on spectrum models that conform to the characteristics of real wave environment, like the JONSWAP spectrum with the Ad-type distribution (Babanin and Solov'yev, 1987; Babanin and Soloviev, 1998). However, the usage of these criterions in real cases was not specified in their research. There is still a lack of a way to introduce arbitrary 2D wave spectra into the criterions, and it is still unknown how effective these criterions are in practical applications. Moreover, in the spectrum models mentioned above, the triggering of MI is a sudden change, depending on whether a certain criterion (a certain combination of the three spectral geometries) is met. In fact, the effect of MI—kurtosis is a quantity that varies gradually and continuously with the change of the geometries, as reflected in the non-Gaussian references. From the perspective of the kurtosis indicator $R_{\mu4}$, the influence of spectral geometries and MI on the possibility of triggering rogue waves can be more intuitively presented.

All the explanations above have been added into the revised manuscript, please see the blue texts in the sections of introduction and discussion.

Comment 2: *What are "additional complicated factors"?*

Response: Please see the Response for Comment 1.

Comment 3: *"...although the range of DS might become narrower as the waves become more developed"*, *please cite some references or show some evidences.*

Response: This phenomenon can be observed in the Alwyn events, and it also has been studied in previous research (Babanin and Soloviev, 1998). According to the $A_d$-type distribution proposed by Babanin and Soloviev (1998) (the reference has been added to the revised manuscript):

$$A_d(f)^{-1} = \int_{-\pi}^{\pi} K(f,\theta)\mathrm{d}\theta \ \text{ and } \ \int_{-\pi}^{\pi} A_d(f)K(f,\theta)d\theta = 1,$$

higher values of $A_d$ correspond to narrower directional distributions. The dependence of the parameter $A_d$ on wave development stage $U/c_m$ at peak frequency $f_m$ is shown in the figure below (which is cited from Fig. 5 in Babanin and Soloviev (1998)):

[Figure]

**Fig. 5.** Dependence of parameter *A* on wave development stage $U/c_m$ at peak frequency $f_m$. Dashed line is the dependence of the parameter *A* on parameter $U/c$ if directional spectra at all frequencies $f$ are taken into account. Level of isotropic directional spectra is shown at the bottom as well as data ranges of the parameterizations of Mitsuyasu *et al.* (1975), Hasselmann *et al.* (1980) and Donelan *et al.* (1985). Triangles show positions of the theoretical spectra (31) with $a = 2.6 \times 10^{-3}$.

Parameter $U/c_m$ represents the wave age, where U is the wind speed and $c_m$ is the wave velocity at peak frequency. Lower values of $U/c_m$ denote more developed wave fields. It is observed in the figure shown above, as parameter $U/c_m$ reduces, larger values of $A_d$ can be found, indicating narrower distribution of wave energy, i.e., narrower DS.

Comment 4: *the latter (the red lines) is several times larger than the former (the bule lines) in all cases of Alwyn, actually. Some explanation on this discrepancy are needed.*

Response: First of all, we need to apologize that the additional HOS simulations performed on the Alwyn_r2–Alwyn_r8 events incorrectly used the spectrum of Alwyn_r1 as initial conditions. Therefore, the black and red lines shown in the Alwyn panels in Fig. 9 appear to be at the same level, whereas blue dashed lines were getting lower and lower through Alwyn_r2–Alwyn_r8, conforming to the smaller and smaller values of $R_{\mu 3}$ and $R_{\mu 4}$ shown in Table 3 (Table 4 in the revised manuscript, see the Response to Comment 5 below). And that's why the red lines are obviously

larger than the blue ones in all the cases of Alwyn. We sincerely thank referee#1 for pointing out our mistakes. The additional HOS simulations were conducted again on all the 8 Alwyn events with the correct initial conditions, and Fig. 9 was redrawn in the revised manuscript. The newly drawn Fig. 9 proves the inactivity of MI in the Alwyn events, and better goodness of the fit of the blue dashed lines to the red lines can be found in the redrawn figure.

Comment 5: *The infinite water depth was adopted for HOSM simulation (line 125) in this study, but was this assumption valid for these sites (of Draupner, Andrea, and Alwyn)?*
Response: Water depth ($d$) at the sites of Draupner, Ekofisk Field, and Alwyn platforms are about 70m, 74m, and 126m (see the newly added column *Depth*(m) in Table 2), and the values of parameter $k_p d$ simulated at the times of the occurrences are listed in Table 3. The values of $k_p d$ shown in Table 3 are all greater than the well-known 1.363, indicating that it was not the water depth that restricted the nonlinear focusing caused by MI in those events (Benjamin and Hasselmann, 1967; Janssen and Onorato, 2007; Whitman, 1974).

A new table (Table 2) was added into the revised manuscript, and an explanation corresponding to this response was added at the end of Sect. 3.4.

Comment 6: *The title is a bit long*
Response: We will seriously consider this comment.

Comment 7: *Typesetting issues:*
Response: The wrong fonts in "Abstract" and "Conclusion and Discussion" have been fixed. The incorrect line spacings in sections 3.x have been fixed too. The changes mentioned above were NOT marked in blue text.

Reference:
Babanin, A. V. and Solov'yev, Y. P.: Parameterization of the width of the angular distribution of wind wave energy at limited fetches, Izv. - Atmos. Ocean Phys., 1987.
Babanin, A. V. and Soloviev, Y. P.: Variability of directional spectra of wind-generated waves, studied by means of wave staff arrays, Mar. Freshw. Res., doi:10.1071/MF96126, 1998.
Barbariol, F., Benetazzo, A., Carniel, S. and Sclavo, M.: Space–Time Wave Extremes: The Role of Metocean Forcings, J. Phys. Oceanogr., 45(7), 1897–1916, doi:10.1175/JPO-D-14-0232.1, 2015.
Barbariol, F., Alves, J.-H. H. G. M., Benetazzo, A., Bergamasco, F., Bertotti, L., Carniel, S., Cavaleri, L., Y. Chao, Y., Chawla, A., Ricchi, A., Sclavo, M. and Tolman, H.: Numerical modeling of space-time wave extremes using WAVEWATCH III, Ocean Dyn., 67(3–4), 535–549, doi:10.1007/s10236-016-1025-0, 2017.
Benjamin, T. B. and Hasselmann, K.: Instability of Periodic Wavetrains in Nonlinear Dispersive Systems [and Discussion], Proc. R. Soc. A Math. Phys. Eng. Sci., 299(1456), 59–76, doi:10.1098/rspa.1967.0123, 1967.
Fedele, F., Brennan, J., Ponce de León, S., Dudley, J. and Dias, F.: Real world ocean rogue waves explained without the modulational instability., Sci. Rep., 6(1), 27715, doi:10.1038/srep27715, 2016.
Janssen, P. and Bidlot, J. J.-R.: On the extension of the freak wave warning system and its verification,

Tech. Memo., (588), 42, 2009.

Janssen, P. a. E. M. and Onorato, M.: The Intermediate Water Depth Limit of the Zakharov Equation and Consequences for Wave Prediction, J. Phys. Oceanogr., 37(10), 2389–2400, doi:10.1175/JPO3128.1, 2007.

Janssen, P. A. E. M.: Shallow-water version of the Freak Wave Warning System, ECMWF Technical Memorandum 813, ECMWF., 2017.

Ribal, A., Babanin, A. V., Young, I., Toffoli, A. and Stiassnie, M.: Recurrent solutions of the Alber equation initialized by Joint North Sea Wave Project spectra, J. Fluid Mech., 719, 314–344, doi:10.1017/jfm.2013.7, 2013.

Whitman, G. B.: Linear and nonlinear waves., , doi:10.4249/scholarpedia.4308, 1974.

Xiao, W., Liu, Y., Wu, G. and Yue, D. K. P.: Rogue wave occurrence and dynamics by direct simulations of nonlinear wave-field evolution, J. Fluid Mech., 720, 357–392, doi:10.1017/jfm.2013.37, 2013.

---

## Author Comment (AC2) · 22 Nov 2020

**Numerical investigation on spectral geometries and their relation to non-Gaussianity in sea states with occurrence of rogue waves: wind-sea dominated events**

Xingjie Jiang[1], Tingting Zhang[1,2], Dalu Gao[1], Daolong Wang[1,2]

[1]First Institute of Oceanography (FIO), Ministry of Natural Resources (MNR), Qingdao, 266061, China
[2]Ocean University of China, Qingdao, 266071, China

*Correspondence to*: Xingjie Jiang (jiangxj@fio.org.cn)

**Abstract.** The occurrence of rogue waves is closely related to the non-Gaussianity of sea states, and the non-Gaussianity is sensitive to the combination of three spectral geometries: wave steepness, bandwidth, and directional spreading. This paper presents a set of non-Gaussianity references that allow quantitative comparison of the non-Gaussianity of sea states with various combinations of the three geometries. In addition, an approach to introduce arbitrary 2D wave spectra into the references is presented, which allows convenient and quantitative investigation of the non-Gaussianity and the corresponding geometries in given sea states. Application in relation to certain rogue waves that occurred in wind-sea dominated sea states showed that the non-Gaussianity of skewness presented high values when those events occurred. However, abnormal values of kurtosis could not be found within the same period, indicating that third-order modulational instabilities were inactive in those events. Quantitative analyses based on the newly presented references revealed that the rogue waves that occurred in wind-sea dominated sea states, and presented extreme height and extreme destructive power, could hardly be formed from the modulational instabilities. This was because of not only the broad energy distribution in terms of direction, but also the broad bandwidth attributable to the developed wind-sea state.

**1 Introduction**

Rogue/freak/extreme waves are highly destructive ocean waves that represent serious threat to various marine activities, e.g., sea voyages, ocean fishing, and oil exploitation. Several physical mechanisms have been proposed to explain the formation of such waves (Kharif et al., 2009; Kharif and Pelinovsky, 2003), including both linear and nonlinear theories. In explaining the occurrence of rogue waves in open seas, it has been suggested that the nonlinear mechanisms that relate to the second- and third-order nonlinear wave–wave interactions appear most reasonable (Fedele, 2008; Fedele and Tayfun, 2009; Janssen, 2003).

The nonlinear wave-energy focusing caused by these nonlinearities can lead to the formation of rogue waves, but it causes the statistics of wave surface elevations to deviate from the Gaussian (normal) distribution, resulting in non-Gaussian sea states (Longuet-Higgins, 1963). Commonly used measures of the non-Gaussianity of sea state are skewness $\mu_3$ and (excess) kurtosis $\mu_4$:

$$\mu_3 = \frac{\langle \eta^3 \rangle}{\langle \eta^2 \rangle^{3/2}}, \mu_4 = \frac{\langle \eta^4 \rangle}{\langle \eta^2 \rangle^2} - 3, \tag{1}$$

where $\eta$ denotes the wave surface elevation and the terms in angled brackets denote statistical averages. It is clear that the skewness is contributed entirely by the second-order nonlinear interactions between bound waves (Fedele and Tayfun, 2009; Janssen, 2009; Tayfun, 1980; Tayfun and Fedele, 2007). The kurtosis comprises a dynamic component ($\mu_4^{free}$) due to third-order quasi-resonant interactions (Janssen, 2003; Mori and Janssen, 2006) between free waves and another bound component ($\mu_4^{bound}$) induced by both second- and third-order bound-wave nonlinearities (Fedele, 2008; Fedele and Tayfun, 2009; Janssen and Bidlot, 2009; Tayfun, 1980; Tayfun and Lo, 1990), which can be written as follows:

$$\mu_3 = \mu_3^{bound}, \mu_4 = \mu_4^{free} + \mu_4^{bound}. \tag{2}$$

The non-Gaussianity of sea state is also sensitive to the geometries of the corresponding wave spectrum, and the relation has been well established using theoretical models, e.g., Janssen (2003,2009) and Fedele and Tayfun (2009), and confirmed by laboratory/numerical experiments, e.g., Onorato et al. (2009a,b), Toffoli et al. (2009), Waseda et al. (2009), and Fedele (2015). Generally, at least three geometries, i.e., wave steepness (hereafter, SP), bandwidth (BW), and directional spreading (DS) can influence skewness, or kurtosis, or both. For example, skewness might be determined by SP and BW (Fedele and Tayfun, 2009); and dynamic kurtosis could be determined by the ratio of SP to BW (Janssen, 2003). Further research, e.g., Fedele (2015), Ribal et al. (2013), and Xiao et al. (2013), highlighted the importance of the spectral geometry of DS. It is concluded that dynamic kurtosis reduces significantly as the directional spreading widens, indicating the third-order quasi-resonant nonlinearities and the associated modulational instabilities (MI) are suppressed.

Through these theoretical and numerical investigations, the trends of change of skewness/kurtosis with SP, BW, and DS have been well studied. For steeper SP and narrower BW and DS, it can be concluded that the deviation from Gaussianity will be greater, resulting in a higher probability of the occurrence of a rogue wave in such a sea state. And there are existed approaches to obtain the indicators of skewness and kurtosis through the three geometries, and some of these indicators can even be applied to operational rogue wave forecast systems, e.g., the ECMWF-IFS WAM (ECMWF, 2016; Janssen, 2017; Janssen and Bidlot, 2009) and the Space–Time Extremes (STE) forecasting included in version 5.16 of the WWIII (Barbariol et al., 2015, 2017; The WAVEWATCH III R Development Group, 2016).

However, those operational non-Gaussian indicators adopted in rogue wave forecast systems are not suitable to describe the non-Gaussianity and its relation to the spectral geometries in real sea states. Those indicators, especially indicators representing dynamic kurtosis, were first obtained from theoretical models derived under the narrowband assumption in an environment with unidirectional wave propagation. When applying them to real sea conditions, specific parameters representing the spectral geometries were selected and undetermined parameters were introduced. Calibrations were undertaken according to the final forecast results, i.e., the exceptional maximum wave/crest height, rather than the skewness/kurtosis observed in real environment. The skewness/kurtosis of wave surface depends on the number of waves involved in the statistics, and the scope of the statistical sea surface and the length of the statistical time can hardly be defined. Therefore, the number of waves is a quantity that cannot be accurately estimated in real wave fields. Furthermore, the number of waves is also an important factor determines the exceptional wave/crest heights forecasted, together with the skewness/kurtosis indicators in the forecast systems. Details of introducing and calibrating the operational non-Gaussianity indicators can be found in Barbariol et al. (2015; 2017) for the WWIII-STE system, and in ECMWF Technical Memorandums (Janssen, 2017; Janssen and Bidlot, 2009) for the ECMWF-IFS system.

[revised manuscript text omitted]

390    Finally, it should be noted that the non-Gaussianity references were calculated with the assumption of infinite depth, but the water depth at the sites of Draupner, Ekofisk Field, and Alwyn platforms are finite as shown in Table 2. According to Table 3, the values of parameter $k_p d$ at the times of the occurrences are all greater than the well-known 1.363, indicating that it was not the water depth that restricted the nonlinear focusing caused by MI in the selected events (Benjamin and Hasselmann, 1967; Janssen and Onorato, 2007; Whitman, 1974).

**395   4    Conclusions and discussion**

In this study, we established a systematic approach that allowed convenient and quantitative comparison of the non-Gaussianity of real-world sea states through the corresponding wave spectra. The newly established approach includes: i) a set of pre-calculated references representing the relation of non-Gaussianity to various combinations of three spectral geometries: SP, BW, and DS, and ii) an approach to introduce arbitrary 2D spectra into the references. The non-Gaussianity

400    references were established based on numerous HOSM simulations, in which various 2D wave spectra were adopted as initial conditions and non-Gaussianity indicators were obtained from the simulated non-Gaussian sea states. The connections between the SP, BW, and DS of the initial spectra and the corresponding non-Gaussianity indicators obtained then constituted the references. We also applied the references to some real rogue wave events. The rogue wave sea states were reproduced using the spectral wave model WWIII, and the 'introducing' approach was adopted to introduce the modelled 2D

405    spectra into the references. Then, analyses focusing on the three spectral geometries and their corresponding non-Gaussianity in the selected sea states were performed.

It was found that all the selected rogue waves occurred in wave fields that were entirely dominated by wind-sea systems, and that waves had become well developed before those events occurred. From the perspective of non-Gaussianity, the sea states

410    all presented greater skewness and less kurtosis when the events occurred, indicating that the selected sea states were dominated by second-order nonlinearities and that the third-order MI were suppressed in such sea states (as confirmed by additional HOSM simulations). From the perspective of spectral geometries, SP was relatively steep when those events occurred, whereas BW and DS were both reasonably broad. According to the non-Gaussianity references, it was the steeper SP that was closely related to the greater non-Gaussianity of skewness, and it was the broader BW and DS that might have

415    suppressed the third-order MI, resulting in less kurtosis in those selected sea states.

It is known that wind waves propagate multi-directionally, while active MI can only be observed in unidirectional wave fields. Therefore, it can be inferred qualitatively that rogue waves occurring in wind-sea states with large DS cannot be formed by the MI. However, narrower DS might be observed at the precise time of occurrence of some rogue events, e.g., the Draupner and Andrea events investigated in this study. The newly proposed non-Gaussianity references provided quantitative support regarding this topic, and it was confirmed that the DS in the selected events was far from the range in which MI could be triggered, even with the narrower DS observed in the Draupner and Andrea events. Furthermore, still based on the quantitative references, it was found that the broad BW observed in developed waves might also have created an unsuitable environment for the generation of MI. In fact, rogue waves with extremely large wave height and extreme destructive power, such as the Draupner and Andrea waves, generally occur in stormy sea states, where the well-developed wave environment could provide energy to support such rogue events. However, owing to the relatively broad BW and DS in such sea states, such giant rogue waves cannot be formed by the third-order quasi-resonant nonlinearities and the associated instabilities.

In comparison with the operational non-Gaussianity indicators adopted in rogue wave forecast systems (see in Sect. 1), the newly proposed non-Gaussianity references were established based on the HOSM simulations, which allow arbitrary spectral width to be involved without the narrowband or unidirectional limitations. In conjunction with the introducing approach, the non-Gaussianity references can be applied to real sea conditions without any undetermined parameters. Furthermore, to avoid the impact of the number of waves on the statistics of skewness/kurtosis, the number of waves was set to be almost the same ($25.5 \times 25.5$) in the initial filed of each HOSM simulation (see Sect. 2.1), and the ratios denoted as $R_{\mu 3}$ and $R_{\mu 4}$ were introduced (see Eq. (8) in Sect. 2.3) to represent the non-Gaussianity obtained under these similar experimental environments.

In this study, HOSM was not the first time been applied to real-world sea states. As mentioned in Sect. 3.3, Fedele et al. (2016) conducted similar HOSM simulations on some rogue wave events and produced similar results about the inactive MI. In fact, HOSM will become very cumbersome if it is applied to every wave spectrum obtained in a time duration of interest. In this study, thanks to the pre-calculated references and the 'introducing' approach, non-Gaussianity indicators $R_{\mu 3}$ and $R_{\mu 4}$ could be obtained very conveniently. Then, the evolution of non-Gaussianity within concerned wave fields and time durations can be presented, as exhibited in Fig. 8.

In addition to illustrating the importance of the factor DS in the estimation of dynamic kurtosis, some of the previously mentioned research also provided criterions containing the three geometries for MI, e.g., parameter $\Pi_2$ in Ribal et al. (2013) and *MBFI* in Xiao et al. (2013). These criterions were derived based on spectrum models that conform to the characteristics of real wave environment, like the JONSWAP frequency spectrum (Hasselmann et al., 1973) with the $A_d$-type distribution (Babanin and Solov'yev, 1987; Babanin and Soloviev, 1998). However, the usage of these criterions in real cases was not

specified in their research. There is still a lack of a way to introduce arbitrary 2D wave spectra into the criterions, and it is still unknown how effective of these criterions are in practical applications. Moreover, in the spectrum models mentioned above, the triggering of MI is a sudden change, depending on whether a certain criterion (a certain combination of the three spectral geometries) is met. In fact, the effect of MI—kurtosis is a quantity that varies gradually and continuously with the change of the geometries, as reflected in the non-Gaussian references. From the perspective of the kurtosis indicator $R_{\mu 4}$, the impact of spectral geometries and MI on the possibility of triggering rogue waves can be more intuitively presented.

With respect to the real ocean, our model is surely an oversimplification. For example, it focuses purely on the nonlinearities between waves, ignoring other physical mechanisms that might influence the non-Gaussianity. Furthermore, it was established based on only unimodal spectral shapes, ignoring bi-modal and even multi-modal shapes, even though such shapes might result in wide BW and DS that could make the sea state less conducive to rogue wave occurrence. Nevertheless, this study provided a new perspective for the study of rogue wave sea conditions, and further research could be undertaken on this basis.

**Author Contribution**

The paper and its methodology were conceptualized and developed by Jiang Xingjie, who also conducted the experiments and data analysis work. The original draft was wrote by Jiang Xingjie, and other co-authors also contributed to preparing, editing, drawing, etc.

**Competing interests**

The authors declare that they have no conflict of interest.

**Acknowledgements**

This work was supported by the National Key Research and Development Program of China (Nos. 2016YFC1401805, 2016YFC1402004). We thank Guillaume Ducrozet and Yves Perignon from the LHEEA of the École Centrale de Nantes and CNRS for their great assistance in helping us understand the HOS method and the use of HOS-ocean. We thank James Buxton MSc from Liwen Bianji, Edanz Group China (www.liwenbianji.cn/ac), for editing the English text of this manuscript.

[revised manuscript text omitted]

---

## Referee Comment (RC3) · Anonymous Referee #1 · 24 Nov 2020

Reply to the author's comments:

I still cannot support this manuscript for the publication. The author's comments are laid out in *italicized* font. My response is given in normal font.

Reviewer's Response:

Overall, there are still expressions (**bold** fonts) that I disagree with in the author's comment and the manuscript.

Comment 1: *This manuscript aims at providing a systematic approach that allows a **convenient** and quantitative comparison of non-Gaussianity of real-world wave fields through the corresponding wave spectra. ... Apparently, the newly proposed approach is **operational** and can be used in more studies related to rogue wave sea states.*

  For whom is your method convenient? Is your method reusable for readers? The computational burden of HOSM is not small, and it is difficult for some readers to perform HOSM simulation similar to yours. If you argue that your method is convenient and operational, the way readers can reuse your results should be provided. For example, your HOSM results corresponding to Fig.3 could be released as a public database or some regression formula relating the kurtosis and skewness to spectral geometries. As a reference, Annenkov & Shrira (2014) showed formula predicting kurtosis from JONSWAP's peakedness and steepness.

Comment 2: *The operational non-Gaussianity indicators, especially dynamic kurtosis, were first obtained from theoretical models derived under the narrowband assumption in an environment with near unidirectional wave propagation. The original expression of those indicators **cannot be applied** to the real sea conditions with broad bandwidth and directional spreading. ... However, calibrations were conducted based on the final forecast results,... the number of waves is a quantity that cannot be accurately estimated in real wave fields. (And that is the "additional complicated factors")*

  I did not see a significant advantage of your method compared to the previous theories. I thought that your statement could be summarized as "better model, better results", but your method using HOSM is still not convincing if comparison to the observation is not provided. Simple methods (like models based on the narrow-band assumption) calibrated well often outperforms complex methods (like HOSM) without calibration. Since ECMWF calibrated their theory with the observation, if anything, their method is more convincing. I admit that the number of waves is a quantity that

cannot be accurately estimated in real wave fields, but it is practically sufficient if the final forecast results can predict the maximum wave/crest height well.

The "additional complicated factors" are associated with statistical calibration. It is possible for me to interpret that your statement denies statistical calibration. However, since the nature of wave in the ocean is very complex and there is no perfect model that treats all physical phenomena, I believe that the calibration is inevitable to treat factors neglected in models statistically. For example, physical factors such as ocean currents, wave breaking neglected in your HOSM might affects the kurtosis. Even more, although the red lines and the blue lines in Fig. 9 are comparable, there are some errors. The error of your method should be statistically calibrated for operational prediction.

The error itself is of interests of readers who have willing to apply your results. Some kinds of error index like RMSE, SI, correlation coefficient and so on should be shown.

In addition, I think the sentence *"The original expression of those indicators **cannot be applied"** in the author's comment is too strong. I agree with the revised sentence "*However, the usage of these criterions in real cases was not specified in their research. There is still a lack of a way to introduce arbitrary 2D wave spectra into the criterions (line 450-451 in the revised manuscript)*", but the previous theories (e.g. Annenkov & Shrira, 2014; Ribal et al., 2013) can be applied if JONSWAP spectra is fitted to spectra output by wave models.

Comment 3-7: I understand your comments. I revised my opinion, and now I think the title is not too long.

In conclusion, if you focus on practical (operational) advantage of your method in comparison with the conventional methods such as that of ECMWF, comparison between your method and the observation should be provided. If the comparison is difficult to do, this study should focus on how to reuse your results for interests of readers as mentioned above.

If you focus on theoretical aspects of your method, and if you would show how to reuse your results, it would be convenient for theorists. Theorists can compare your results to previous theories (e.g. Annenkov & Shrira, 2014; Ribal et al., 2013). Since HOSM M=2 includes some part of four-wave interactions and HOSM M=3 exhibits Class II instability associated with five-wave interactions (Fujimoto et al., 2019; Fujimoto & Waseda, 2016), HOSM M=3 might not necessarily show results same as the previous theories. The relationship between HOSM and the Zakharov equation is not trivial, and difference between kurtosis calculated by HOSM and the Zakharov equation might be of interest for theorists.

---

## Referee Comment (RC4) · Anonymous Referee #2 · 24 Nov 2020

**Review**

on the manuscript "Numerical investigation on spectral geometries and their relation to non-Gaussianity in sea states with occurrence of rogue waves: wind-sea dominated events" by Xingjie Jiang, Tingting Zhang, Dalu Gao, Daolong Wang, submitted for publication in NHESS.

This work is dedicated to the study of the rogue wave phenomenon, aiming at solving the problem of forecasting the occurrence of such waves. In the article, the set of spectral parameters, the wave steepness, the frequency bandwidth and the directional spreading, is used for determining the probabilistic properties of the wave field in terms of skewness and kurtosis. The relation between the spectral and statistical parameters is estimated using the direct numerical simulations within the HOSM for irregular waves with the JONSWAP spectrum and the $\cos^2$ spreading function. The obtained relation is applied to the sea conditions observed during the occurrence of widely known rogue waves, such as the New Year Wave, the Andrea Wave and some others. The sea state conditions were reproduced by the WWIII model; they turn out to be characterized by broad frequency and directional spectra. The authors make use of the possibility to control the order of nonlinearity of the HOSM to show the dominant contribution of the second-order nonlinearity to the third statistical moment (skewness); and that the fourth statistical moment (kurtosis) is small and the 4-wave interactions do not lead to its increase. They conclude that in these particular sea states the effect of the modulation instability is negligible.

The idea of the work is generally understood and reasonable; in the literature there have already been some attempts to tackle the problem following similar approaches (some references are present in the manuscript). However, there is a long list of uncertainties on the way to the final outcomes; not all the conclusions sound convincing. On the other hand, the work represents a new serious attempt to reconstruct the realistic wave conditions within numerical simulations. Unfortunately, the work is lacking in discussion and comparison with previously reported results on similar problems, even though the publications are mentioned. In particular, the BFI parameter is not mentioned in the work at all, although it seems to be a most promising characteristic of the extremality of sea states. The in-situ extreme events considered in the article have already been analyzed earlier, but the new results are not discussed against the already reported. Such a discussion should be added to the revised version of the manuscript before it is accepted for publication. The text is generally not easy-to-read due to the numerous notations, which are not always obvious. Some particular issues listed below should be clarified as well.

1. Line 228. "*It should be noted that $R_{\mu4}$ could take a negative value with some combinations of ($\varepsilon$, $\gamma$, $\Theta$), which could be attributable to two factors.*" I have doubts about the negative values of kurtosis for rather small directional spreading. It could be seen from Fig. 3(c,d) that the negative values of kurtosis are observed for small wave steepness (looking at the figure, I cannot estimate the kurtosis value for larger $\varepsilon$). I think this may be due to insufficient statistics of waves for the initial conditions, when a Gaussian field is constructed. If the number of random waves is insufficient, the kurtosis will be less than zero (see negative dynamic kurtosis in Fig. 5a in [Slunyaev A. V. Effects of coherent dynamics of stochastic deep-water waves. Phys.Rev.E, 101, 062214, 2020] ). Note however that the total kurtosis is positive, though the directional spreading is large and the steepness is relatively small). If the subsequent evolution of the waves of small steepness is practically linear, then the wave statistics may remain almost unchanged.
Secondly, such negative values might be a result of the high-frequency cut-off. The similar effect of kurtosis underestimation is discussed in the recent preprint [Kokina, T. Dias, F. Influence of the Wave Spectrum on Statistical Wave Properties. Preprints 2020, 2020110421]. Please indicate

the rate of the energy loss during the simulation (within the last 170 periods) caused by the wave breaking suppression.

2. In the numerical experiments by HOSM (section 2.1), the spatial sampling was about 10 points per a wavelength. To the best of my knowledge, it is not sufficient to describe accurately the wave features, what may lead to artifacts of processing. A greater resolution was used in, particular, [Xiao, W., Liu, Y., Wu, G. and Yue, D. K. P. Rogue wave occurrence and dynamics by direct simulations of nonlinear wavefield evolution, J. Fluid Mech., 720, 357–392, 2013]. The issue of sufficient spatial resolution was particularly considered in [Slunyaev A., Kokorina A, Account of occasional wave breaking in numerical simulations of irregular water waves in the focus of the rogue wave problem. Water Waves, 2, 243-262, 2020], where about 20 points per wave was used as the minimum.

3. The statistical moments are calculated by averaging over a remarkably long period of 170 after the start of the simulation. It is well-known that the transition of modulationally unstable wave systems to a quasi-stationary state takes a couple of tens wave periods (e.g., [Shemer L. and Sergeeva A. An experimental study of spatial evolution of statistical parameters in a unidirectional narrow-banded random wave field. J. Geophys. Res., 114, C01015, 2009; Shemer L., Sergeeva A., Slunyaev A. Applicability of envelope model equations for simulation of narrow-spectrum unidirectional random field evolution: experimental validation. Phys. Fluids, 22, 016601(1-9), 2010] ). Therefore the averaging over 170 wave periods will most likely completely hide the effects of the modulational instability. It is well seen in Fig. 9 (Alwyn-r2, for example) that during the first 10-20 $T_p$ both the skewness and kurtosis increase rapidly and then oscillate around some steady-state value. For steeper waves this effect will be more pronounced. It may be more reasonable to consider a shorter time period (and to simulate a greater number of realizations). Anyway, it would be instructive if the maximum attained values of skewness and kurtosis for the given spectral parameters are plotted in e.g. Fig. 8,9 in line with the averaged ones.

4. In Sec. 3.2 the values of the skewness and kurtosis which characterize the Draupner wave and the Andrea wave seem to be rather different from the results of a similar analysis performed in [Fedele, F., Brennan, J., Ponce de León, S., Dudley, J. and Dias, F. Real world ocean rogue waves explained without the modulational instability., Sci. Rep., 2016]. Please discuss possible reason of this disagreement.

Some other remarks:

Line 114. What is "*a relevant parameter n=4*"?

Line 121. Please clarify what "*pseudo-spectra*" means? The same question about "*pseudo-spectral space*" in Table 1.

Line 138-140. In Eq. (4) should be *β* instead of *B*.

Line 143. The writing $\cos^2 x$ should be probably used in Eq. (5).

Line 185. It should be $\mu_3$ in the expression for the skewness, Eq. (8).

Line 215. "*It can be seen in Fig. 3c and 3d that within the range where Θ is extremely narrow (e.g., Θ ≤ 20°), $R_{\mu4}$ decreases markedly as Θ widens; when Θ is beyond the extremely narrow range, the value of $R_{\mu4}$ reaches a much lower level and decreases markedly more slowly in comparison with the situation when Θ widens within the extremely narrow range.*" The surfaces in Figs. 3 are probably not the best way to present the data, since the non-monotonic change of kurtosis is hardly seen. I suggest using contour plots or something similar (the top-view with a color coding, etc.).

Line 221 "*…in a normal sea state (Annenkov and Shrira, 2014)*".Do you mean "Gaussian sea state" or "typical sea state"?

Line 231. "*Negative $R_{\mu4}$ values represent sea states with less possibility of finding rogue waves; thus, they would not influence identification of MI-triggering combinations.*" Please explain what you mean.

Line 256. The subscripts *i* of the characteristics $\varepsilon \ \gamma \ \Theta$ are not defined.

Line 301. "*It can be seen from Fig. 7 and Table 3 that the parameters indicating SP are very similar at the times of occurrence of the selected events, i.e., they are almost all within the range of 0.035–0.040.*" The meaning of the sentence is not clear. Please paraphrase.

Table 2. Please add to Table 2 the information about $T_p$ and the local water depth.

Figure 2. The blue colors look similar, therefore the lines are poorly readable. Please change.

Figure 3. Please, swap the figures c) and d) to be consistent with a) and b).

Figure 4. Please, correct the colorbar (depth, meter). The scale of the depths does not allow to estimate the bathymetry in the vicinity of the locations of measurements. Please indicate the depths of the measurement locations.

Figure 7. The absence of the red curves in the upper panel looks confusing. You could use different line widths or line styles to show the curves which coincide. A new undefined notation *Spr* appears in the bottom figure. Please change the scale of the left vertical axis – the red lines are poorly read.

Figure 8. The figure is impossible to read. I can see only two horizontal lines corresponding to the averaged values. I suggest the authors to use different colors and line widths.

Figure 9. I would expect that the blue dashed lines show the averaged values of the dependences shown with the red curves. However, the plots are inconsistent with this. Could you please explain the relation between the curves in the figure. Please, give the values of $B_{\mu3}$ and $B_{\mu4}$ on the graph.

In Figures 9, 10 the axis tick labels are too dense and even overlap. Please, improve.

---

## Author Comment (AC3) · 26 Nov 2020

Please see the attached files.

Please also note the supplement to this comment:
https://nhess.copernicus.org/preprints/nhess-2020-342/nhess-2020-342-AC3-supplement.zip

———————————————————

2020-342, 2020.

---

## Author Comment (AC4) · 9 Dec 2020

We sincerely thank referee#2 for the valuable feedback that we have used to improve the quality of our manuscript. The referee's comments are laid out in italicized font and the comments have been numbered in the authors' responses. Our responses are given in normal font and changes/additions to the manuscript according to referee#2 are given in red text (blue texts are revisions according to referee#1, some of the revisions can also be considered as responses to referee#2's comments). The authors' responses and the revised manuscript can be found in the attachment.

[Figure]

Sincerely Yours, Dr. Jiang Xingjie

Please also note the supplement to this comment:
https://nhess.copernicus.org/preprints/nhess-2020-342/nhess-2020-342-AC4-supplement.zip